# V-JEPA: Latent Video Prediction for Visual Representation Learning

## Abstract

This paper shows that the masked-modelling principle driving the success of large foundational language models can be effectively applied to video by making predictions in latent space. We introduce V-JEPA, a method for self-supervised learning from video that predicts masked spatio-temporal regions in a learned representation space. Our latent video prediction strategy produces visual features that can be applied to various downstream image and video tasks without adaption of the model's parameters, achieving $82.1\%$ on Kinetics-400 and $71.2\%$ on Something-Something-v2, surpassing the previous best video models by $+4$ and $+10$ points respectively when using a frozen evaluation protocol. We also demonstrate the benefit of video pretraining compared to image pretraining for tasks involving motion understanding, where V-JEPA outperforms the largest state-of-the-art image models, DINOv2 and OpenCLIP. Finally, V-JEPA trained only on videos achieves $77.9\%$ on ImageNet classification without any image fine-tuning, surpassing the previous best video model by $+6$ points top-1.

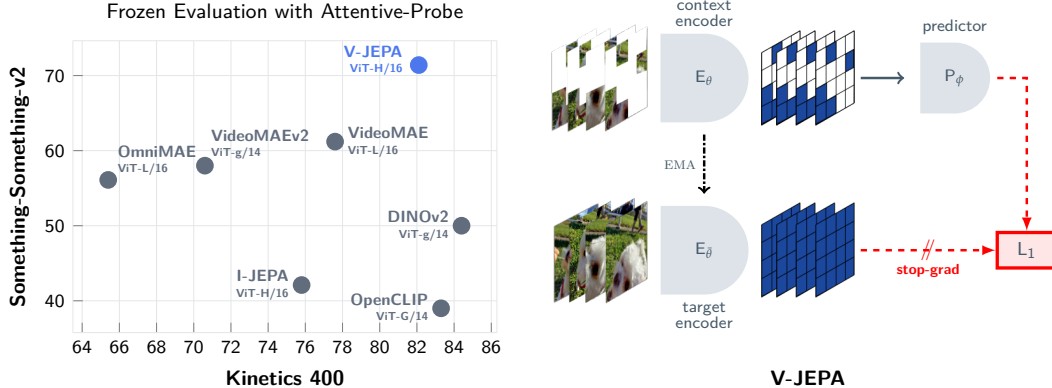

Figure 1: **(Left)** V-JEPA pretrained on video learns visual representations that perform well on motion-based tasks (Something-Something-v2) and appearance-based tasks (Kinetics 400) by training a single cross-attention layer on top of the frozen encoder backbone. **(Right)** V-JEPA trains a visual encoder by predicting masked spatio-temporal regions in a learned latent space.

## 1 Introduction

Several cognitive theories assert that a fundamental learning mechanism in biological systems is the adaptation of an internal model to predict missing input information (Rao & Ballard, 1999; Friston, 2005). This self-supervised principle of learning by filling-in-the-blanks has fueled the recent success of large foundation language models (Devlin et al., 2018; Mann et al., 2020; Touvron et al., 2023), which are trained to predict masked words in a large text corpus. However, it remains an open question how to best instantiate this learning principle with sensory data such as video.

The recently proposed Joint-Embedding Predictive Architecture (JEPA) (LeCun, 2022; Assran et al., 2023) demonstrated that one could learn representations of images that perform well off-the-shelf

(without fine-tuning) by predicting masked image regions in a learned representation space. Compared to other methods for masked image modelling (He et al., 2021; Tong et al., 2022; Feichtenhofer et al., 2022), which predict low-level visual tokens or pixels, JEPAs make predictions in a learned latent space, where unpredictable pixel-level details can be eliminated, thereby leading the model to learn more semantic features (Assran et al., 2023).

In this work, we study the problem of self-supervised representation learning from video data and extend the JEPA-based learning principle to video. We propose to train a visual encoder by predicting missing regions of a spatio-temporally masked video in a learned representation space. We refer to this approach as V-JEPA: Video-based Joint-Embedding Predictive Architecture (cf. Figure 1). Empirically, we show that training a V-JEPA model on 2 million publicly available videos from academic datasets leads to strong off-the-shelf performance on a diverse set of image and video tasks, including SomethingSomething-v2 (SSv2) (Goyal et al., 2017), AVA (Gu et al., 2018), ImageNet (Russakovsky et al., 2015), and Kinetics (Kay et al., 2017).

On visual tasks that require a semantic temporal understanding of a scene, such as action classification on SSv2 or action localization on AVA, we find V-JEPA to provide a significant boost in frozen evaluation over the largest state-of-the-art image and video models, such as VideoMAE (Tong et al., 2022; Wang et al., 2023), DINOv2 (Oquab et al., 2023), and OpenCLIP (Radford et al., 2021; Cherti et al., 2023). Specifically, we achieve $71.2\%$ top-1 frozen evaluation on Something-Something-v2, surpassing VideoMAE, DINOv2 and OpenCLIP by $+10$, $+21$, and $+32$ points respectively (cf. Figure 1). On tasks that require good image-level features, such as Kinetics and ImageNet, we observe that V-JEPA is an effective pretraining strategy. We obtain the first unsupervised video model to achieve $82.1\%$ top-1 frozen evaluation accuracy on Kinetics, without any fine-tuning, outperforming the previous best video model, VideoMAE, by $+4.2$ points. Similarly, V-JEPA obtains $77.9\%$ top-1 on ImageNet, without any image fine-tuning.

Finally, through a series of ablations, we highlight the importance of the pretraining data distribution on the model's downstream performance. Specifically, we find that increasing the video dataset size continues to improve performance, even with a fixed computation budget, hinting at a promising path to further improve video foundation models.

## 2 BACKGROUND

How can we train machines to perceive the visual world? Towards answering this long-standing question, various families of approaches have been proposed for visual representation learning from static images and videos.

**Weakly supervised learning.** One family of approaches trains a visual encoder to predict the representations of text captions often found accompanying images from the Web, as in CLIP (Radford et al., 2021). The largest open source CLIP model to date, numbering 2B parameters and trained on over 2B web-scraped images (Cherti et al., 2023), demonstrates impressive performance on a wide range of downstream image and video tasks. Notably, this is achieved using only the light-weight adaptation of task-specific heads, also referred to as frozen-evaluation, and does not require expensive end-to-end fine-tuning of the pretrained model. This family of approaches has been extended to video data by leveraging closed captioning, often computed from an ASR transcription of audio data accompanying internet videos. For instance, VideoBERT (Sun et al., 2019; Xu et al., 2021) trains a video encoder to predict masked spans in the textual closed captions. Similarly, VideoCLIP (Xu et al., 2021) trains a video encoder to predict the representation of video captions computed by a text encoder. Follow-up work such as MERLOT Reserve (Zellers et al., 2022), VATT (Akbari et al., 2021), and InternVideo (Wang et al., 2022) extended VideoCLIP by adding a self-supervised audio or video loss. We compare our work with the latter two models and show that V-JEPA surpasses their performances on downstream tasks while using no text supervision.

**Self-supervised learning from Images.** Self-supervised methods aim to learn representations without the need for human-annotated training data. Invariance-based pretraining, for instance, trains a visual encoder to be invariant to hand-crafted image transformations (Chen et al., 2020; Caron et al., 2020), however, such methods still require a significant amount of task-specific inductive bias (e.g., through specifying the set of transformations), which can limit their applicabil-

ity (Assran et al., 2022a). Denoising auto-encoders present an alternative approach for learning representations by reconstructing a corrupted input (Vincent et al., 2008). Masked Autoencoders (MAEs) (He et al., 2021) use this principle to train a visual encoder-decoder that predicts the pixels of image patches that are masked at the input. When finetuned, MAEs achieve state-of-the-art performance on image recognition tasks. However, the representations produced by MAEs require involved adaptation procedures and fall short of the representations produced by invariance-based methods in frozen-evaluations (Assran et al., 2022b). Other works (Baevski et al., 2022b; Assran et al., 2023; Baevski et al., 2022a) have explored masked image modelling tasks while predicting in a representation space rather than raw pixels. One such method in particular, I-JEPA (Assran et al., 2023), demonstrated strong performance on downstream tasks using only frozen evaluations, without requiring end-to-end fine-tuning.

The DINOv2 method of Oquab et al. (2023) combines an invariance-based loss with a masked-image modelling loss and is currently the most competitive instantiation of self-supervised learning with (static) images, scaled to a model with over 1.1B parameters trained on a curated dataset of 142M images. Notably, DINOv2 demonstrated the feasibility of matching or surpassing CLIP in frozen evaluations while using no textual supervision during pretraining. However, since it is only trained on static images, we do not expect DINOv2 representations to be able to capture information about motion dynamics in video. We will show that by pretraining on video, V-JEPA outperforms DINOv2 on video tasks that require an understanding of the underlying video motion.

**Self-supervised learning from Videos.** Self-supervised learning has also been explored with video. For instance, Feichtenhofer et al. (2021) explores the slow-feature assumption (Wiskott & Sejnowski, 2002), which encourages the learning of invariant representations across time in a video. MC-JEPA (Bardes et al., 2023) trains a ConvNet to predict the representation of the next frame using a hierarchical architecture. VITO (Parthasarathy et al., 2022) demonstrates that self-supervised learning from video can lead to strong representations for image segmentation. VideoMAE (Tong et al., 2022; Feichtenhofer et al., 2022) and VideoMAEv2 (Wang et al., 2023) directly extend the masked autoencoder approach to spatio-temporal volumes by training an encoder-decoder model to predict masked spatio-temporal voxels. Li et al. (2023) train a model using a masked-modelling loss that is computed in the frozen representation space of a pretrained CLIP encoder. Contrary to that work, V-JEPA predicts in a representation space that is learned online during training. Omni-iMAE (Girdhar et al., 2023) simultaneously trains a visual encoder on both images and video. In an orthogonal direction, Ryali et al. (2023) demonstrate the benefit of a hierarchical transformer architecture for masked autoencoding, while Gupta et al. (2023) explore masked autoencoding using a frame-level encoder and a cross-attention based video decoder. The MAE-based approaches induce minimal inductive biases during pretraining and exhibit strong performance when fine-tuning on downstream visual tasks, however, we will empirically demonstrate that V-JEPA leads to better off-the-shelf representations.

**Evaluation protocols.** Toward the goal of learning a generalist model that can solve a wide range of tasks, it is common to measure the performance of large-scale image and video foundation models under a frozen-evaluation procedure, where the pretrained network is frozen, and a small set of task-specific parameters are optimized on top of the frozen backbone (Chen et al., 2020; Oquab et al., 2023; Yuan et al., 2023). These task-specific parameters are often specified as a linear layer processing the output of the frozen backbone, or a cross-attention layer, which pools the output feature map into a single feature vector, followed by a linear layer. The latter procedure is often referred to as attentive pooling (Huang et al., 2022). Frozen evaluation has been studied in large-scale evaluations of image and video models, such as in Oquab et al. (2023) and Yuan et al. (2023).

## 3 METHODOLOGY

**Joint-Embedding Predictive Architecture.** The main idea behind a JEPA (LeCun, 2022) is to learn by using one part of an input (context), to predict another part of the input (target), in an abstract latent space. The basic architecture is made up of three networks: a context encoder, $E_\theta(\cdot)$, which computes the representation of the context, a target encoder, $E_{\bar{\theta}}(\cdot)$, which computes the representation of the target, and a predictor, $P_\phi(\cdot, \cdot)$, which predicts the target representation from

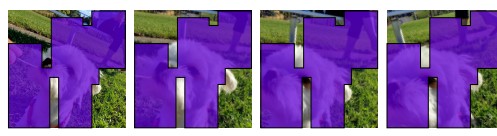 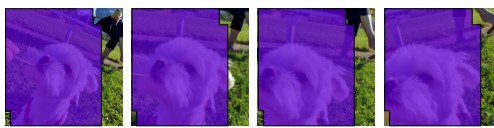

(a) **Short-range mask.** Sample 8 blocks of scale 0.15 with aspect ratio in the range $(0.75, 1.5)$, and take their union.

(b) **Long-range mask.** Sample 2 blocks of scale 0.7 with aspect ratio in the range $(0.75, 1.5)$, and take their union.

Figure 2: **3D Multi-Block Masking.** We leverage two 3D Multi-Block masking strategies during pretraining: short-range masks and long-range masks, which lead the model to capture different types of features in a video.

the context representation given information about the relative transformation between them. All three networks are trained by minimizing the error between the predictor and target encoder outputs.

We instantiate the architecture for video with V-JEPA by using masking; see Figure 1 (right). The context-encoder and predictor networks process a masked video and output a prediction about the content of the masked regions. This prediction is then regressed via an $L_1$ loss to the output of the target encoder, which processes the full (unmasked) video.

**Architecture.** As previously mentioned, we use a standard Vision Transformer (ViT) (Dosovitskiy et al., 2020) as our default video backbone, which processes a 1D sequence of tokens, each of dimension $d$, and outputs a $d$-dimensional embedding vector for each token. The *context encoder* is a regular ViT network (ViT-L or ViT-H) consisting of standard transformer blocks with joint space-time attention. The *target encoder* is also a regular ViT network, initialized identically to the context encoder. Finally, the *predictor* is a narrow ViT implemented using 12 transformer blocks with an embedding dimension of 384. For simplicity, we keep the number of self-attention heads in the predictor equal to that of the context encoder.

**Input.** Since ViT networks process a 1D sequence of tokens, we must first convert an input video clip into a 1D token sequence. To do so, we apply a 3D convolution comprising $d$ filters of size $2 \times 16 \times 16$ with a temporal stride of 2 and a spatial stride of 16, resulting in a tensor of shape $8 \times 14 \times 14 \times d$. Next we add absolute 3D sin-cos positional embeddings to the spatio-temporal feature map and flatten it, resulting in a 1D token sequence of shape $1568 \times d$. This process is demonstrated in Figure 4 in Appendix A. Unless stated otherwise, during during pretraining, we always randomly sample a clip of 16 frames from each input video with a temporal stride of 4 between sampled frames. An input video clip therefore covers 64 frames in total, or roughly 2 seconds of a given video running at 30 frames per second. We then resize the video's spatial dimensions to $224 \times 224$, resulting in an overall shape of $16 \times 224 \times 224 \times 3$ for the entire clip.

**3D Multi-Block Masking.** We use a simple 3D extension of the block masking strategy employed for images (Bao et al., 2021). Given a video, we sample several (possibly overlapping) spatially continuous blocks with various aspect ratios and take their union to construct a single mask. This spatial mask is then repeated across the entire temporal dimension. Masking a large continuous block that covers the full temporal dimension limits information leakage due to the spatial and temporal redundancy of videos, and results in a harder prediction task (Tong et al., 2022).

In V-JEPA, we consider two types of masks: short-range masks, where we take the union of 8 randomly sampled target blocks with a spatial scale of 0.15, and long-range masks, where we take the union of 2 randomly sampled target blocks with a spatial scale of 0.7. In both cases, the aspect ratio for all sampled blocks is randomly chosen in the range $(0.75, 1.5)$. Given that both short-range and long-range masks are produced by sampling many blocks and taking their union, the result is an average masking ratio of $\sim 90\%$ with our masking hyper-parameters, meaning that the context encoder only needs to process $\sim 10\%$ of the video. Figure 2 illustrates our masking strategy, which we refer to as **3D Multi-Block**. Extensive ablations on the 3D Multi-Block Masking hyper-parameters are provided in Appendix C.3.

**Patch-Level Loss.** To compute the V-JEPA loss in each iteration, we sample both a video clip, and a video mask. We denote a video clip represented as a 1D token sequence of length $L = 1568$ by $\boldsymbol{x}_L = (x_1, \ldots, x_L)$. Similarly, given a mask of $M < L$ patches, leaving $N = L - M$ patches

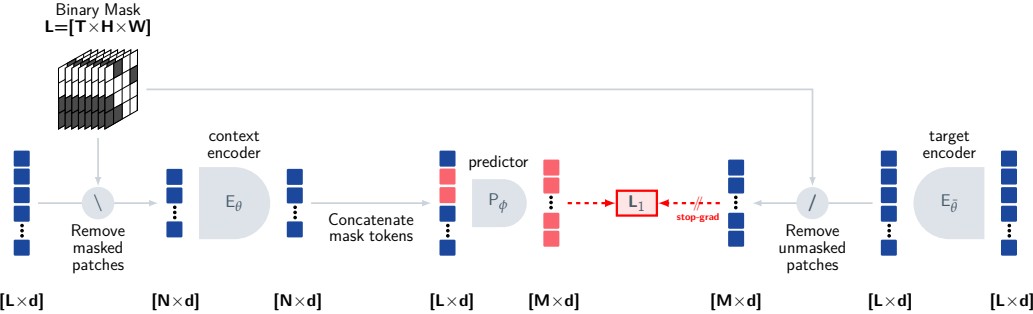

Figure 3: **V-JEPA** training operates on a video clip flattened into a sequence of tokens. The sequence is masked and fed to the context-encoder. The output of the context-encoder is then concatenated with a set of learnable mask tokens, indicating the spatio-temporal positions of the masked patches, and fed to the predictor network, which outputs an embedding vector for each mask token. To compute the prediction targets, the target encoder processes the complete (unmasked) token sequence. The output of the predictor is then regressed to the corresponding output tokens of the target encoder using an $L_1$ loss.

unmasked, we denote the indices of masked patches by $(i_1, \ldots, i_M)$ and its complement (the indices of unmasked patches) by $(j_1, \ldots, j_N)$.

*Computing the context representations.* To compute the V-JEPA loss, we first produce the context representations by masking the video clip and feeding it into the context encoder; we denote the masked video by $\boldsymbol{x}_N = (x_{j_1}, \ldots, x_{j_N})$. Applying the context encoder $E_\theta(\cdot)$ to the masked clip gives a sequence of patch representations, denoted as $\boldsymbol{z}_N = E_\theta(\boldsymbol{x}_N) = (z_{j_1}, \ldots, z_{j_N})$.

*Predicting the target regions.* Next, the V-JEPA predictor network $P_\phi(\cdot, \cdot)$ takes as input the tokens produced by the context encoder and predicts the missing regions in the video clip, which are specified by a set of learnable mask tokens. Specifically, the mask tokens are parameterized as the sum of a shared learnable vector and an absolute 3D sin-cos positional embedding, denoted by $\boldsymbol{m}_M = (m_{i_1}, \ldots, m_{i_M})$. The output of the predictor is thus given by, $\hat{\boldsymbol{s}}_M = P_\phi(\boldsymbol{z}_N, \boldsymbol{m}_M) = (\hat{s}_{i_1}, \ldots, \hat{s}_{i_M})$, corresponding to a $d$-dimensional output for each of the $M$ masked patches.

*Computing the target representations.* Finally to compute the prediction targets, the entire unmasked video clip is processed by the target encoder to obtain a set of target representations, denoted by $\boldsymbol{s}_L = E_{\bar{\theta}}(\boldsymbol{x}_L) = (s_1, \ldots, s_L)$. The V-JEPA loss is now computed as

$$\text{Loss} = \frac{1}{M} \sum_{k \in (i_1, \ldots, i_M)} \|\hat{s}_k - s_k\|_1, \tag{1}$$

which is simply the average $L_1$ distance between the output of the predictor and the target encoder. We then compute a gradient update with respect to the parameters of the context encoder, $\theta$, and the predictor, $\phi$, and subsequently update the parameters of the target encoder, $\bar{\theta}$, as an exponential moving average of the context encoder weights (Polyak average). A detailed diagram of the V-JEPA procedure is provided in Figure 3.

**Multi-Mask Prediction.** To increase the efficiency of V-JEPA, we use a multi-masking strategy (Caron et al., 2020; Baevski et al., 2022a), which enables us to amortize the cost of the target computation. As previously mentioned, for a given video clip, we sample 2 different masks, short-range and long-range. While we need to forward propagate the context encoder and predictor separately for each mask, we only need to compute the target representation once.

**Collapse Prevention Mechanism.** Representation collapse, wherein the encoder produces a constant output regardless of the input, is a trivial solution of our loss function. By adding a stop-gradient at the output of the target encoder and updating its weights as an exponential moving average of the context encoder, we are able to avoid this uninformative solution. This collapse prevention mechanism has previously been employed by image-based methods for self-supervised learning (Chen & He, 2020; Grill et al., 2020; Baevski et al., 2022b). A theoretical motivation for its effectiveness was proposed in Grill et al. (2020) for the BYOL method. We provide a simple adaptation of this analysis for our $L_1$ loss in Appendix D.

## 4 EXPERIMENTS

### 4.1 EXPERIMENTAL SETTING

**Pretraining.**   We pretrain V-JEPA on a combination of publicly available academic datasets. Specially, we combine Kinetics-400/600/700 (Kay et al., 2017), HowTo100M (Miech et al., 2019), and Something-Something-v2 (Goyal et al., 2017), and remove any videos appearing in the validation sets of Kinetics-400/600/700 and Something-Something-v2. Through this process, we obtain a dataset for self-supervised pretraining containing approximately 2 millions videos, which we refer to as VideoMix2M. All V-JEPA models are trained for 90,000 iterations on VideoMix2M with a batch size of 3072 for the ViT-L/16 and ViT-H/16 models, and a batch size of 2400 for the ViT-H/16$_{384}$ model. Note that this training schedule is an order of magnitude shorter than previous state-of-the-art video and image models; see Appendix C.2. Pretraining details are reported in Appendix A.

**Evaluation Datasets.**   We evaluate models on both video tasks and static-image tasks. On video tasks, we use a subset of the VideoGLUE benchmark (Yuan et al., 2023) to test for various capabilities; specifically, we investigate action recognition on Kinetics-400 (K400) (Kay et al., 2017), motion classification on Something-Something-v2 (SSv2) (Goyal et al., 2017), and action localization on AVA (Gu et al., 2018). Action classification on Kinetics evaluates the content understanding of the model, since many action classes in the dataset can be inferred from the presence of certain objects in the video (Sevilla-Lara et al., 2021). Motion classification on Something-Something-v2 evaluates the temporal understanding of the model, since action classes in the dataset are decoupled from the appearance/presence of specific objects in the video (Goyal et al., 2017). Finally, action localization on AVA evaluates the ability of the model to understand and localize motions in the video. For static-image tasks, we explore object recognition on ImageNet (Russakovsky et al., 2015), scene classification on Places205 (Zhou et al., 2014), and fine-grained recognition on iNaturalist 2021 (Van Horn et al., 2018).

**Evaluation Protocol.**   In contrast to specialist models that only focus on one task, our goal is to learn a generalist video model that can be quickly adapted to a wide variety of tasks. We therefore explore a frozen-evaluation protocol where the video encoder is kept fixed after pretraining and lightweight task-specialized probes are learned for each downstream tasks. For classification, we use the attentive probing protocol (Chen et al., 2022; Yuan et al., 2023) where a simple cross-attention module with a learnable query token and a linear classifier is learned on top of the frozen encoder. For spatio-temporal action detection, we extend the practice used for 2D segmentation (Bardes et al., 2022) and learn a linear head on top of the frozen backbone. More details regarding the evaluation implementation are provided in Appendix B. Fine-tuning on a given tasks usually improves a model's accuracy while reducing its generality; see Section 4.4 for fine-tuning evaluations.

**Baselines.**   We report the performance of V-JEPA and various self-supervised and weakly-supervised baselines trained on static images and videos to allow the research community to compare V-JEPA with previous works. Our self-supervised video model baselines consist of Video-MAE (Tong et al., 2022) and VideoMAEv2 (Wang et al., 2023), which train exclusively on videos, and OmniMAE, which combines a self-supervised loss on static images with a self-supervised loss on videos. Our weakly-supervised video model baselines consist of InternVideo (Wang et al., 2022) and VATT (Akbari et al., 2021), which combine a weakly-supervised contrastive text loss with a self-supervised video loss. We also evaluate baselines trained exclusively on static images. Our image model baselines consist of OpenCLIP (Cherti et al., 2023), which uses a weakly-supervised contrastive text loss, DINOv2 (Oquab et al., 2023), which uses a self-supervised view-invariance loss, and I-JEPA (Assran et al., 2023), which uses a self-supervised masked image modelling loss. The I-JEPA method is particularly relevant for our analysis, since it is also based on the concept of a joint-embedding predictive architecture.

### 4.2 EVALUATION ON VIDEO TASKS

This section uses a frozen evaluation protocol to measure the ability of a model to produce video embeddings that excel at many downstream tasks. While task-specific end-to-end usually improves a model's accuracy, it also reduces its generality. We note that specialist models using end-to-end fine-

Table 1: **Frozen Evaluation on Video Datasets.** We compare V-JEPA with various image and video baselines in frozen evaluation with an attentive probe. Models are pretrained using either image, video, text or a combination of those modalities. V-JEPA outperforms previous video model across all tasks. On tasks requiring motion understanding, V-JEPA outperforms image-pretrained models by over $+21$ points. Results shown in gray are reported from the attentive probe evaluation in Yuan et al. (2023). All the other results are obtained with our attentive probe evaluation pipeline.

| Method | Pretrain Data | Arch. | Image | Video | Text | K400 | SSv2 | AVA |
|---|---|---|---|---|---|---|---|---|
| *Methods pretrained on Images* | | | | | | | | |
| I-JEPA | IN1K | ViT-H/16$_{448}$ | ✓ | | | 74.5 | 42.1 | 16.3 |
| OpenCLIP | LAION | ViT-G/14 | ✓ | | ✓ | 83.3 | 39.0 | 23.2 |
| DINOv2 | LVD-142M | ViT-g/14 | ✓ | | | **84.4** | 50.0 | 24.3 |
| *Methods pretrained on Videos* | | | | | | | | |
| InternVideo | UnlabeledHybrid | - | | ✓ | ✓ | 73.7 | 60.3 | 19.6 |
| VATT | HowTo100M | - | | ✓ | ✓ | 75.1 | 58.7 | 22.9 |
| OmniMAE | IN1K + SSv2 | ViT-L/16 | ✓ | ✓ | | 65.4 | 56.1 | 14.4 |
| Hiera | K400 | Hiera-H | | ✓ | | 73.3 | 60.7 | 17.5 |
| VideoMAE | K400 | ViT-L/16 | | ✓ | | 77.9 | 61.2 | 21.6 |
| VideoMAEv2 | UnlabeledHybrid | ViT-g/14 | | ✓ | | 70.6 | 58.0 | 12.9 |
| V-JEPA | K400 | ViT-L/16 | | ✓ | | 78.7 | 61.7 | 25.0 |
| | VideoMix2M | ViT-L/16 | | ✓ | | 79.1 | 67.0 | 25.6 |
| | | ViT-H/16 | | ✓ | | 81.7 | 69.1 | **25.8** |
| | | ViT-H/16$_{384}$ | | ✓ | | **82.1** | **71.2** | 25.0 |

tuning still achieve absolute state-of-art on those tasks. InternVideo (Wang et al., 2022) achieves an accuracy of 91.1 and 77.2 on K400 and SSv2 respectively, while Hiera (Ryali et al., 2023) achieves a score 43.3 on AVA.

**Comparing with video models.** Table 1 reports the performance of pretrained video models on downstream video tasks under the attentive probing protocol. On K400, V-JEPA achieves a score of 82.1% and surpasses the previous best video model by $+4.2$ points; on SSv2, V-JEPA achieves a score of 71.2% and surpasses the previous best video model by $+10$ points; and on AVA, V-JEPA achieves a score of 25.8 mAP and surpasses the previous best video model by $+2.9$ mAP.

**Comparing with image models.** Table 1 also demonstrates the importance of video pretraining (as opposed to pretraining on static images) for learning general visual representations. Recall that the Something-Something-v2 benchmark requires a fine-grained understanding of video motion, as appearance features are not informative with respect to the action labels. We observe that V-JEPA provides a major improvement (over $+21$ points) on this task compared to strong image-baselines such as DINOv2 and OpenCLIP, which are trained on large-scale image datasets. We hypothesize that self-supervised pretraining from videos allows us to model dynamic concepts that are not possible to learn from static image dataset. Similarly, on the action-localization task, we observe that the V-JEPA models outperform image-based pretraining.

However, it is worth noting that static image models still learn strong static visual features compared to models that are pretrained on videos. By evaluating the DINOv2 model with an attentive probe on K400, we report a new state-of-the-art baseline on this task. Specifically, while DINOv2 (Oquab et al., 2023) previously reported 78.4% on K400 with a linear probe, we improve their frozen evaluation to 84.4% by using an attentive probe. It has previously been reported in Sevilla-Lara et al. (2021) that motion information is less important than appearance-based cues to solve this task, and our results further support this observation.

**Controlling for the pretraining distribution.** We pretrain a ViT-L/16 using V-JEPA on the K400 dataset (comprising 240K videos) using the regular pretraining hyperparameters described in Section 4.1. We compare this model to a VideoMAE ViT-L/16 also pretrained on K400. VideoMAE uses similar masked video modelling loss, but in pixel space. V-JEPA has a consistent advantage over VideoMAE with an improvement of $+0.7$, $+0.5$ and $+3.4$ points on K400, SSv2 and AvA. Furthermore, V-JEPA demonstrates a favourable scaling behavior; by pretraining on VideoMix2M (comprising 2M videos), the performance is boosted by $+0.5$, $+5.3$ and $+0.6$ points, respectively. We do not observe a similar performance improvement when evaluating a VideoMAEv2 model pretrained on the UnlabeledHybrid dataset (comprising 1.35M videos).

Table 2: **Frozen Evaluation on Image Datasets**. We compare V-JEPA with image and video baselines in frozen evaluation with an attentive probe. Models are pretrained using either image, video, text or a combination of those modalities. V-JEPA outperforms previous self-supervised video models across all tasks. While image pretrained models show an advantage on downstream image tasks, V-JEPA significantly reduces the gap between image and video models. Scores are obtained with our attentive probing evaluation pipeline.

| Method | Pretrain Data | Arch. | Image | Video | Text | IN1K | Places | iNat21 |
|---|---|---|---|---|---|---|---|---|
| *Methods pretrained on Images* | | | | | | | | |
| I-JEPA | IN1K | ViT-H/16$_{448}$ | ✓ | | | 81.1 | 63.5 | 80.3 |
| OpenCLIP | LAION-2B | ViT-G/14 | ✓ | | ✓ | 85.3 | **70.2** | 83.6 |
| DINOv2 | LVD-140M | ViT-g/14 | ✓ | | | **86.2** | 68.4 | **88.8** |
| *Methods pretrained on Videos* | | | | | | | | |
| OmniMAE | IN1K+ SSv2 | ViT-L/16 | ✓ | ✓ | | 75.1 | 59.8 | 66.1 |
| VideoMAE | K400 | ViT-L/16 | | ✓ | | 71.9 | 60.0 | 65.8 |
| VideoMAEv2 | UnlabeledHybrid | ViT-g/14 | | ✓ | | 71.4 | 60.6 | 68.3 |
| V-JEPA | VideoMix2M | ViT-L/16 | | ✓ | | 73.7 | 60.5 | 67.4 |
| | | ViT-H/16 | | ✓ | | 75.3 | 61.1 | 68.3 |
| | | ViT-H/16$_{384}$ | | ✓ | | **77.4** | **61.8** | **73.4** |

## 4.3 EVALUATION ON IMAGE TASKS

Table 2 investigates performance on image tasks with an attentive probe. In particular V-JEPA achieves a score of $77.4\%$ on ImageNet using a one-layer attentive probe, which can be further improved to **77**.**9**$\%$ using a two-layer attentive probe. While, V-JEPA outperforms models pretrained on video, such as VideoMAE, the largest image models still have a clear advantage for image tasks. We hypothesize that the distributions of the video datasets are too constrained and lack the visual diversity of internet-scale pretraining data used by the images models. We further investigate this question in Section 4.5 by pretraining image models on static frames from video datasets.

## 4.4 FINETUNING

In Table 3 we evaluate a V-JEPA model by finetuning (separately) on K400 and SSv2, and comparing with VideoMAEv2 (Wang et al., 2023) and VideoMAEv1 (Tong et al., 2022) models. The pretraining time per iteration is measured with a batch-size of 16 on an A100 using our codebase for V-JEPA, and the official codebases for VideoMAEv2 and VideoMAEv1. V-JEPA outperforms VideoMAEv1 by $+0.1\%$ on K400 and $+0.5\%$ on SSv2, when using 16 frames clip for fine-tuning. While V-JEPA time per iteration is higher than that of VideoMAEv1, V-JEPA model is trained for fewer iterations, thus our result is obtained in 60% of the VideoMAEv1 training time. The VideoMAEv2 model improves on VideoMAEv1 by training a ViT-g/14 model for an order of magnitude more iterations. Efficiency is maintained by using a shallower 4-layer decoder and employing decoder masking. This VideoMAEv2 ViT-g/14 model outperforms our V-JEPA model in finetuning by $+0.5\%$ on K400 and $+1.7\%$ on SSv2; however, this result requires training $6.5\times$ longer than V-JEPA. Additionally, we note that by increasing the number of frames to 32 instead of 16, we can further improve the V-JEPA score to 75.9 on SSv2.

Table 3: **Finetuning results.** We evaluate a V-JEPA model with the finetuning protocol on the K400 and SSv2 datasets using 16 frames per clip and multi-view fusion (5×3 or 2×3) for inference. The **#Samples Seen** entry corresponds to the number of video clips processed during pretraining, which is larger than the size of the pretraining dataset for multi-epoch training. The pretraining time per iteration is measured with a batch-size of 16 on an A100 using our codebase for V-JEPA, and the official codebases for VideoMAEv2 and VideoMAEv1. We report the VideoMAEv2 results without instruction-turning for consistency with the other approaches.

| Method | Arch. | Data | #Samples Seen | Time/Itr (s) | K400 (16×5×3) | SSv2 (16×2×3) | Slowdown |
|---|---|---|---|---|---|---|---|
| V-JEPA | ViT-H/16$_{384}$ | VideoMix2M | 210M | 2.225 | 86.7 | 75.3 | **1.0**× |
| VideoMAEv1 | ViT-L/16 | K400\|SSv2 | 380M\|410M | 1.924 | 85.2 | 74.3 | 1.56×\|1.69× |
| | ViT-H/16 | K400\|SSv2 | 380M\|410M | 2.018 | 86.6 | 74.8 | 1.65×\|1.76× |
| VideoMAEv2 | ViT-H/16 | Un.Hybrid | 1600M | 1.867 | 86.9 | 76.8 | 6.4× |
| | ViT-g/14 | Un.Hybrid | 1600M | 1.904 | 87.2 | 77.0 | 6.5× |

Table 4: **Ablating Pretraining Distribution**

(a) **Image vs. Video Pretraining**. We compare I-JEPA pretrained on IN1K to I-JEPA pretrained on K400 for 400 epochs, and V-JEPA pretrained on K400 for 300 epochs. When controlling for the pretraining distribution, V-JEPA outperforms its image-only counterpart.

| Method | Arch. | Pretraining Data | IN1K |
|--------|-------|------------------|------|
| I-JEPA | ViT-H/16 | IN1K | 78.0 |
| I-JEPA | ViT-H/16 | K400 | 50.4 |
| V-JEPA | ViT-L/16 | K400 | 70.4 |

(b) **Video Pretraining Data Distribution**. Even with *a fixed computational budget*, V-JEPA performance with a ViT-L/16 network continues to increase on SSv2 and IN1k as we increase the scale of the pretraining dataset.

| Dataset | Size ∼ | SSv2 | IN1K |
|---------|--------|------|------|
| K400 | 240K | 61.7 | 70.4 |
| K400/600/700 | 650K | 63.7 | 72.9 |
| VideoMix2M | 2M | 67.1 | 73.7 |

## 4.5 ABLATIONS

**Image versus video pretraining.**  Video data offer rich temporal supervision that can reveal properties such as 3D-geometry, object permanence, solidity/rigidity, affordances, etc. However, Section 4.3 indicates that the best performing models on static image tasks use only image data during pretraining. We hypothesize that this observation is largely due to the visual diversity of static image datasets, which tend to be well aligned with the considered image downstream tasks.

To test this hypothesis in a controlled ablation, we compare video pretraining with V-JEPA to image pretraining with I-JEPA (Assran et al., 2023). The I-JEPA model is also based on a joint-embedding predictive architecture and relies on the same learning principle as V-JEPA, but is applied to static images, where the goal is to predict the representations of missing 2D blocks. We take the official implementation of I-JEPA and first pretrain a ViT-H/16 on ImageNet at resolution $224 \times 224$ to obtain a baseline. Next, we take a randomly initialized model and pretrain it with I-JEPA on the K400 video dataset at resolution $224 \times 224$ by processing individual frames as static images. Finally, we take a randomly initialized ViT-L/16 model, and pretrain it with V-JEPA on the K400 video dataset at resolution $224 \times 224$.

We report frozen evaluation on ImageNet in Table 4a. When controlling for the pretraining data distribution, we see that V-JEPA outperforms I-JEPA on ImageNet-1K classification by approximately $+20$ points, despite using a significantly smaller visual encoder. This suggests that the distribution shift between video and image datasets is an important factor in establishing the differences between image pretraining and video pretraining. This result suggests that simply dedicating more effort towards improving and scaling the distribution of video data, as is already common practice with image datasets (Oquab et al., 2023), may provide a path for producing state-of-the-art image features with video models.

**Impact of pretraining dataset scale.**  To further study the impact of the video pretraining data distribution, we explore the impact of data scale on V-JEPA pretraining *with a fixed computational budget* in Table 4b. First, we pretrain V-JEPA on roughly 200K videos, corresponding to K400, next we pretrain V-JEPA on roughly 650K videos, corresponding to a combination of K400/600/700, and finally we pretrain V-JEPA on roughly 2M videos, corresponding to a combination K400/600/700 and other academic video datasets. Crucially, we keep the number of pretraining iterations *fixed* across the three runs. As show in Table 4b, the performance of V-JEPA improves monotonically with data scale on both motion-based tasks (SSv2) and appearance-based tasks (IN1k).

## 5 DISCUSSION

We introduced V-JEPA, a self-supervised approach for learning representations from video, which can be applied to various downstream image and video tasks without adaption of the model parameters. V-JEPA outperforms previous video representation learning approaches in frozen evaluation on action recognition, spatio-temporal action detection, and image recognition tasks. Additionally, we show that pretraining V-JEPA on videos is particularly effective for solving downstream tasks that require motion understanding, while approaches trained only on internet scale image datasets fall short. In future work, we will explore how scaling and improving the diversity of the pretraining video distribution could be used to improve video foundation models.

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

Table 5: **Pretraining hyper-parameters for V-JEPA.**

| Hyper-parameter | ViT-L/16$_{224}$ | ViT-H/16$_{224}$ | ViT-H/16$_{384}$ |
|---|---|---|---|
| *data* | | | |
| datasets | VideoMix2M | VideoMix2M | VideoMix2M |
| resolution | 224 | 224 | 384 |
| num_frames | 16 | 16 | 16 |
| temporal_stride | 4 | 4 | 4 |
| horizontal_flip | true | true | true |
| random_resize_scale | (0.3, 1.0) | (0.3, 1.0) | (0.3, 1.0) |
| random_resize_aspect_ratio | (0.75, 1.33) | (0.75, 1.33) | (0.75, 1.33) |
| *masking* | | | |
| block_aspect_ratio | (0.75, 1.5) | (0.75, 1.5) | (0.75, 1.5) |
| shortrange_mask_num_blocks | 8 | 8 | 8 |
| shortrange_mask_spatial_scale | 0.15 | 0.15 | 0.15 |
| longrange_mask_num_blocks | 2 | 2 | 2 |
| longrange_mask_spatial_scale | 0.7 | 0.7 | 0.7 |
| *optimization* | | | |
| batch_size | 3072 | 3072 | 2400 |
| total_number_of_iterations | 90000 | 90000 | 90000 |
| warmup_iterations_fraction | 0.15 | 0.15 | 0.15 |
| lr | 6.25e-4 | $6.25\times10^{-4}$ | $6.25\times10^{-4}$ |
| start_lr | $2\times10^{-4}$ | $2\times10^{-4}$ | $2\times10^{-4}$ |
| final_lr | $1\times10^{-6}$ | $1\times10^{-6}$ | $1\times10^{-6}$ |
| start_momentum | 0.998 | 0.998 | 0.998 |
| final_momentum | 1.0 | 1.0 | 1.0 |
| start_weight_decay | 0.04 | 0.04 | 0.04 |
| final_weight_decay | 0.4 | 0.4 | 0.4 |
| scheduler_scale_factor | 1.25 | 1.25 | 1.25 |
| *architecture* | | | |
| patch_size | 16 | 16 | 16 |
| tubelet_size | 2 | 2 | 2 |
| pred_depth | 12 | 12 | 12 |
| pred_embed_dim | 384 | 384 | 384 |
| *hardware* | | | |
| dtype | bfloat16 | bfloat16 | bfloat16 |
| accelerator | A100 80G | A100 80G | A100 80G |

## A  PRETRAINING DETAILS

In section, we report V-JEPA pretraining details. Table 5 summarizes the main hyperparameters used during pretraining.

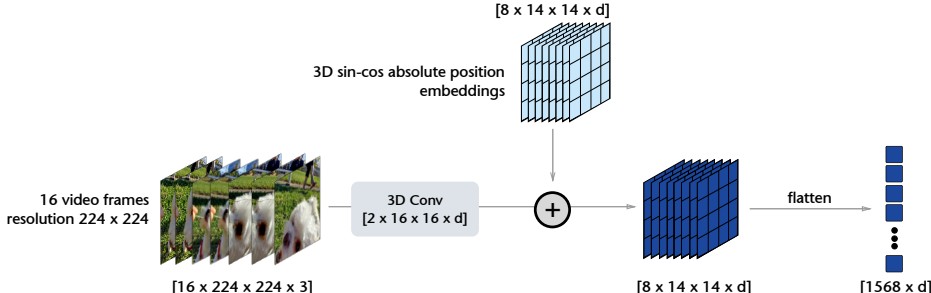

Figure 4: **V-JEPA** training operates on a video clip flattened into a sequence of tokens. To convert a video clip of size $16 \times 224 \times 224 \times 3$ into a 1D token sequence, we apply a 3D convolution comprising $d$ filters of size $2 \times 16 \times 16$ with a temporal stride of 2 and a spatial stride of 16, resulting in a tensor of shape $8 \times 14 \times 14 \times d$. Next we add absolute 3D sin-cos positional embeddings to the spatio-temporal feature map and flatten it, resulting in a 1D token sequence of shape $1568 \times d$.

Table 6: **Frozen Evaluation hyper-parameters.**

| Hyper-parameter | K400 | SSv2 | IN1K | Place205 | iNat21 |
|---|---|---|---|---|---|
| *data* | | | | | |
| num_clips | 8 | 1 | N.A. | N.A. | N.A. |
| num_frames | 16 | 16 | N.A. | N.A. | N.A. |
| temporal_stride | 4 | 4 | N.A. | N.A. | N.A. |
| horizontal_flip | true | true | true | true | true |
| random_resize_scale | (0.08, 1.0) | (0.08, 1.0) | (0.08, 1.0) | (0.08, 1.0) | (0.08, 1.0) |
| random_resize_aspect_ratio | (0.75, 1.33) | (0.75, 1.33) | (0.75, 1.33) | (0.75, 1.33) | (0.75, 1.33) |
| auto_augment | false | false | true | false | false |
| *optimization* | | | | | |
| batch_size | 256 | 256 | 512 | 256 | 256 |
| epochs | 20 | 20 | 20 | 20 | 20 |
| lr | 1e-3 | 1e-3 | 1e-3 | 1e-3 | 1e-3 |
| final_lr | 0 | 0 | 0 | 0 | 0 |
| weight_decay | 0.01 | 0.01 | 0.01 | 0.01 | 0.01 |

**Architectures.** We use Vision Transformer (Dosovitskiy et al., 2020) (ViT) architectures for the context-encoder and target-encoder. We train three V-JEPA encoders: a ViT-L/16$_{224}$, a ViT-H/16$_{224}$ and a ViT-H/16$_{384}$. All three encoders take as input a short video clip of 16 frames with a temporal stride of 4 between consecutive frames. The subscripts, $224$ and $384$, indicate the spatial resolution of the video clip. V-JEPA flattens the video clip into a sequence of non-overlapping spatio-temporal patches of size $16 \times 16 \times 2$ (see Figure 4). For all three models, the predictor is designed as a narrow ViT architecture, consisting of 12 transformer blocks with an embedding dimension of 384. For simplicity, we keep the number of self-attention heads in the predictor equal to that of the backbone used for the context-encoder/target-encoder. V-JEPA is pretrained *without* using a `[cls]` token.

**Optimization.** We use AdamW (Loshchilov & Hutter, 2017) to optimize the context-encoder and predictor weights. The ViT-L/16$_{224}$ and ViT-H/16$_{224}$ models use a batch size of 3072 while the ViT-H/16$_{384}$ uses a batch size of 2400. Models are trained for a total of 90,000 iterations. The learning rate is linearly increased from $2 \times 10^{-4}$ to $6.25 \times 10^{-4}$ during the first $12,000$ iterations of pretraining, and decayed to $10^{-6}$ following a cosine schedule. Weight-decay is also linearly increased from 0.04 to 0.4 throughout pretraining. The target-encoder weights are initialized identically to the context-encoder, and subsequently updated as an exponential moving average (EMA) (Tarvainen & Valpola, 2017) of the context-encoder weights using a momentum value which starts at 0.998 and is linearly increased to 1.0 during training (Caron et al., 2021; Assran et al., 2022b). We scale all hyper-parameter schedules 25% beyond the actual training schedule. Specifically, the learning rate schedule, weight-decay schedule, and EMA schedule are computed assuming a training length of 112,500 iterations, even though we only train our model for 90,000 iterations. We found the last 25% of the default scheduler period to update hyper-parameters too aggressively, and simply truncating the schedulers improved performance.

**Masking.** As described in Section 3, we propose a 3D Multi-Block masking strategy. We use two type of masks: short-range masks, where we take the union of 8 randomly sampled target blocks with a spatial scale of 0.15, and long-range masks, where we take the union of 2 randomly sampled target blocks with a spatial scale of 0.7. In both cases, the aspect ratio for all sampled blocks is randomly chosen in the range (0.75, 1.5).

# B EVALUATION DETAILS

## B.1 FROZEN CLASSIFICATION

**Attentive Probing.** Given an input video, $\boldsymbol{x}_L$, the V-JEPA target encoder $E_{\bar{\theta}}(\cdot)$ outputs a sequence of $L$ tokens, $E_{\bar{\theta}}(\boldsymbol{x}_L) = (s_1, \ldots, s_L)$, where $s_i \in \mathbb{R}^d$. To pool this sequence of tokens into a single feature vector, we apply a lightweight cross-attention using a learnable query token (Chen et al.,

2022). Specifically, our attentive pooling performs the following computation:

$$\sum_{i=1}^{L} \frac{\exp(q^\top \mathbf{W_k} s_i)}{\sum_j \exp(q^\top \mathbf{W_k} s_j)} \mathbf{W_v} s_i,$$

where $\mathbf{W_k}, \mathbf{W_v} \in \mathbb{R}^{\mathbf{d \times d}}$ are the key and value matrices, and $q \in \mathbb{R}^d$ is a learnable query token. The output of the attentive pooler is then fed into a standard linear classifier, and its parameters are jointly learned with that of the linear classifier for the downstream task, while the encoder parameters are kept frozen. Note that, in practice, we actually use an attentive pooler with 12 heads, each of dimension $d/12$. In Appendix C we show that both V-JEPA and baselines benefit from the attentive probing protocol.

**Optimization.**  For all the tasks, we use AdamW optimizer with a cosine scheduler (no warmup) that decays the learning rate from $0.001$ to $0$. We use a fixed weight-decay of $0.01$ and apply simple data augmentations (random resized crops and horizontal flips) during training of the attentive probe, except on ImageNet, where we apply AutoAugment (Dogus Cubuk et al., 2019). Table 6 reports the hyperparameters used for the different evaluation tasks.

**Extension to multiple clips.**  On Kinetics-400 frozen evaluation, our attentive probe takes 8 video clips as input to increase the temporal coverage of the video. Specifically, we first divide a video in 8 equal-length temporal segments and sample 1 clip at random per segment. The video encoder $E_{\bar{\theta}}$ processes each clip separately and produces a clip-level feature map. The feature maps for each clip are then concatenated together and fed to the attentive probe.

**Application of video models to images.**  To evaluate the video models on image tasks, we simply duplicate input images to generate still video clips of 16 frames. We perform this duplication operation simply for convenience in evaluation of the video models, however we find this step to be unnecessary in general. Given a video tokenizer implemented as a 3D-conv with a temporal stride of 2, it is sufficient to simply duplicate the image into a 2 frame video clip. This would result in the same number of input tokens as that produced by a static image model with a 2D-conv tokenizer.

**Application of image models to videos.**  To evaluate image models such as DINOv2 and Open-CLIP on video tasks, we simply process each frame independently with the image encoder to produce a frame-level feature map. The feature maps for each frame are then concatenated and fed to the attentive probe, just as we do with the clip-level feature maps when evaluating video models.

### B.2 FROZEN DETECTION

We evaluate our model on the AVA (Gu et al., 2018) spatio-temporal localization of human actions dataset, containing 211k training and 57k validation video segments. We follow the experimental protocol of (Feichtenhofer et al., 2021), and use precomputed masks from a pretrained Faster-RCNN adapted to videos, which uses a ResNeXt-101-FPN backbone and is pretrained on ImageNet and COCO. We train a linear classifier on top of the *frozen* V-JEPA features to classify the extracted regions of interest and report mean Average Precision (mAP) on the 60 most common classes. Hyperparameters are provided in Table 7. Our frozen features are obtained by concatenating the last layer of the transformer encoder with three intermediate layers. We use a batch size of 64 and pretrain for 30 epochs with AdamW using a learning rate of 0.0001 with 2 epochs of warmup and a weight decay of 0.05.

### B.3 FINETUNING

We evaluate in Appendix 4.4 our V-JEPA ViT-H/16$_{384}$ model on Kinetics-400 and Something-Something v2 in the finetuning setting. Following Tong et al. (2022), we finetune a linear layer on top of our model, using a layer decay schema and mixup as the data augmentation pipeline. We provide all hyper-parameters for both K400 and SSv2 in Table 8.

Table 7: **Frozen Detection hyper-parameters.**

| Hyper-parameter | ViT-L/16 | ViT-H/16 |
|---|---|---|
| out_layers | [18, 20, 22, 24] | [26, 28, 30, 32] |
| batch_size | 64 | 64 |
| epochs | 30 | 30 |
| opt | AdamW | AdamW |
| opt_eps | 0.00000001 | 0.00000001 |
| momentum | 0.9 | 0.9 |
| weight_decay | 0.05 | 0.05 |
| lr | 0.0001 | 0.0001 |
| warmup_lr | 0.000001 | 0.000001 |
| min_lr | 0.000001 | 0.000001 |
| warmup_epochs | 2 | 2 |
| warmup_steps | 1 | 1 |

Table 8: **Finetuning Evaluation hyper-parameters.**

| Hyper-parameter | K400 | SSv2 |
|---|---|---|
| *data* | | |
| num_segments | 1 | 1 |
| num_frames | 16 | 32 |
| sampling_rate | 4 | 4 |
| resolution | 384 | 384 |
| *model* | | |
| model_name | ViT-H | ViT-H |
| tubelet_size | 2 | 2 |
| drop_path | 0.2 | 0.2 |
| head_drop_rate | 0.5 | 0.5 |
| *optimization* | | |
| batch_size | 64 | 64 |
| epochs | 35 | 20 |
| opt | adamw | adamw |
| opt_eps | 0.00000001 | 0.00000001 |
| momentum | 0.9 | 0.9 |
| weight_decay | 0.05 | 0.05 |
| lr | 0.0003 | 0.0001 |
| layer_decay | 0.8 | 0.8 |
| warmup_lr | 0.00000001 | 0.00000001 |
| min_lr | 0.000001 | 0.000001 |
| warmup_epochs | 5 | 5 |
| warmup_steps | 1 | 1 |
| *augmentations* | | |
| color_jitter | 0.4 | 0.4 |
| num_sample | 2 | 2 |
| aa | rand-m7-n4-mstd0.5-inc1 | rand-m7-n4-mstd0.5-inc1 |
| smoothing | 0.1 | 0.1 |
| train_interpolation | bicubic | bicubic |
| test_num_segment | 5 | 2 |
| test_num_crop | 3 | 3 |
| *erase* | | |
| prob | 0.25 | 0.25 |
| mode | pixel | pixel |
| count | 1 | 1 |
| split | False | False |
| *mixup* | | |
| mixup | 0.8 | 0.8 |
| cutmix | 1.0 | 1.0 |
| mixup_prob | 1.0 | 1.0 |
| mixup_switch_prob | 0.5 | 0.5 |
| mixup_mode | batch | batch |

## C  EXTRA RESULTS

### C.1  FROZEN EVALUATION.

Table 9: **Linear vs. Attentive Probe Evaluation for V-JEPA and VideoMAE.** We evaluate the effect of linear (Lin.) and attentive (Att.) probing when adapting V-JEPA to the K400 and SSv2 tasks. V-JEPA and VideoMAE benefit from using a non-linear attentive probe. Specifically, using an attentive probe with V-JEPA leads to an improvement of +22 points on K400 and +17 points on SSv2.

| | | K400 | | SSv2 | |
|---|---|---|---|---|---|
| Method | Arch. | Lin. | Att. | Lin. | Att. |
| VideoMAE | ViT-L/16 | 52.5 | 77.6 | 41.3 | 61.2 |
| V-JEPA | ViT-L/16 | 56.7 | **79.1** | 50.1 | **67.1** |

Table 10: **Linear vs. Attentive Probe Evaluation for DINOv2 and OpenCLIP.** We evaluate the effect of linear (Lin.) and attentive probing (Att.) when adapting DINOv2 and OpenCLIP. Image-baselines benefit from using an attentive probing strategy. Results shown in  gray  are reported from the linear probe evaluation in Oquab et al. (2023).

| | | K400 | | SSv2 | | IN1K | | Place205 | | iNat21 | |
|---|---|---|---|---|---|---|---|---|---|---|---|
| Method | Arch. | Lin. | Att. | Lin. | Att. | Lin. | Att. | Lin. | Att. | Lin. | Att. |
| DINOv2 | ViT-g/14 | 78.4 | 84.4 | 38.3 | 50.0 | 86.5 | 86.2 | 67.5 | 68.4 | 85.7 | 88.8 |
| OpenCLIP | ViT-G/14 | 78.3 | 83.3 | 35.8 | 39.0 | 86.2 | 85.3 | 69.8 | 70.2 | 76.0 | 83.6 |

**Linear vs. Attentive probe**   We compare the effect of using an attentive versus a linear probe when adapting a pretrained model to various downstream tasks. Table 9 shows that V-JEPA and VideoMAE benefit from using a non-linear attentive probe on the K400 and SSv2 downstream tasks. In particular, using an attentive probe with V-JEPA leads to an improvement of +22 points on K400 and +17 points on SSv2. Additionally, Table 10 shows that attentive probing leads to better performance on average for DINOv2 and OpenCLIP models. Since attentive probing improves the performance of all models, we use it as our default evaluation protocol.

Table 11: **Temporal Coverage on Kinetics-400.** We evaluate the effect of temporal coverage on K400. We train an attentive probe on K400 using either 1 clip ($\approx$ 2 seconds of a video) or 8 clips ($\approx$ 16 seconds of a video). To sample $N$ clips, we first divide a video in $N$ equal-length temporal segments and sample one clip at random per segment. The video encoder processes each clip in parallel and all the encoder output tokens are concatenated at the input of the attentive probe. Increasing the temporal coverage from 1 clip per video to 8 clips significantly improves the performance for both our VideoMAE baseline and V-JEPA. Specifically, using 8 clips leads to an improvement of +6.2 points on K400 with a V-JEPA ViT-H/16$_{384}$.

| Method | Arch. | 1 Clip | 8 Clips |
|---|---|---|---|
| VideoMAE | ViT-L/16 | 69.4 | 77.6 |
| V-JEPA | ViT-L/16 | 72.3 | 79.1 |
| | ViT-H/16$_{384}$ | **75.8** | **82.0** |

**Temporal coverage on Kinetics-400.**   We examine the impact of changing the temporal coverage of a model during downstream evaluation on K400 action classification. In Table 11, we evaluate VideoMAE and V-JEPA models using an attentive probe with access to either the feature map of 1 clip randomly sampled from the video, or the concatenated feature map of 8 clips randomly sampled from the video. To sample 8 clips from a video, we first divide the video into 8 equal length temporal segments, and sample 1 clip at random from each segment. A single clip corresponds to $\approx$ 2 seconds of a video on average, while 8 clips correspond to $\approx$ 16 seconds. The video encoders processes each clip separately to produce a clip-level feature map, which are then concatenated at the input to the attentive probe.

Increasing the temporal coverage from 1 clip per video to 8 clips improves the performance of both V-JEPA and VideoMAE on K400 action classification. Specifically, using 8 clips leads to an improvement of +6.2 points on K400 with a V-JEPA ViT-H/16$_{384}$. We therefore use the 8 clip attentive probing setup as our default evaluation pipeline on K400 for all video and image

models. While we would expect multi-clip evaluation to be helpful for other downstream video action classification tasks, we still only sample one clip when training an attentive probe on SSv2, as videos from that dataset are only 2 to 4 seconds long on average.

## C.2 Sample Efficiency of Pretraining

We compare the sample efficiency of pretraining various state-of-the-art image and video models. Specifically, we look at the number of samples (image or video clips) processed by the network during pretraining, which is larger than the size of the pretraining dataset for multi-epoch training. Notably, our results with V-JEPA are obtained while processing an order of magnitude fewer samples than previous methods, and notably two orders of magnitude fewer samples than OpenCLIP. We believe that further investment towards improving the video pretraining data distribution could lead to substantial gains in downstream image and video tasks.

Table 12: **Sample efficiency.** We compare the sample efficiency of pretraining various state-of-the-art image and video models. The **#Samples Seen** entry corresponds to the number of samples (image or video clips) processed by the network during pretraining, which is larger than the size of the pretraining dataset for multi-epoch training. The V-JEPA results in this paper are obtained while processing an order of magnitude fewer samples than previous methods.

| Method | Arch. | Data | #Samples Seen |
|---|---|---|---|
| OpenCLIP | ViT-G/14 | LAION-2B | 39000M |
| DINOv2 | ViT-g/14 | LVD 142M | 1900M |
| VideoMAEv2 | ViT-g/14 | UnlabeledHybrid | 1600M |
| V-JEPA | ViT-H/16$_{384}$ | VideoMix2M | 210M |

## C.3 Masking Strategy

An important component of the V-JEPA pretraining strategy is the 3D clip masking strategy. In this section, we detail 26 ablation experiments exploring different masks. For all the experiments, we pretrain a ViT-B/16 pretrained on K400. Figure 5 presents a summary of those results.

Recall that each video mask is constructed by sampling several (possibly overlapping) blocks and taking their union. These spatial multi-block masks are then repeated along the temporal dimension to create a 3D Multi-Block mask. Figure 5c shows the effect of changing the spatial and temporal masking ratio. Figure 5b ablates the number of sampled blocks used to construct the masks given a fixed effective masking ratio of 90%. Finally, in Figure 5a we examine our multi-masking strategy and find that sampling two masks for each clip (long-range and short-range) to be more effective than sampling just a single mask for each clip. By default we sample masks that remove roughly 90% of the frame and extend along the entire temporal dimension of the clip.

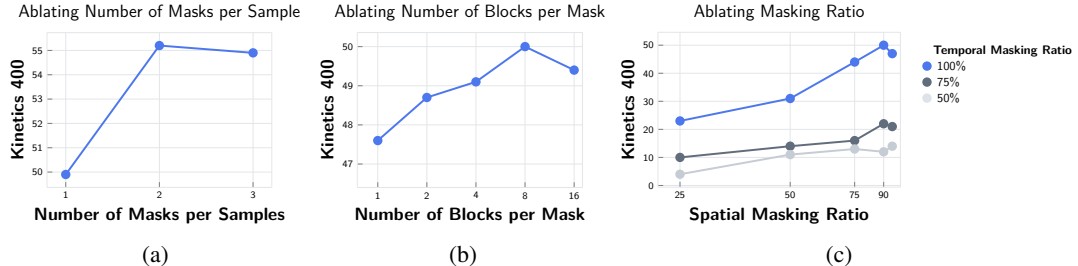

(a)        (b)        (c)

Figure 5: **Masking Strategy Ablation.** Evaluating a linear probe on a ViT-B/16 pretrained with V-JEPA on K400 under various 3D Multi-Block masking settings. We examine the impact of **(a)** sampling several masks per video, **(b)** varying the number of blocks in a mask, and **(c)** varying the average spatial and temporal masking ratio. A temporal masking ratio of 100% extends the spatial mask across all the frames in the clip. We find it important to maintain a high spatial and temporal masking ratio during pretraining.

In Table 13, we explore different average spatial and temporal masking ratio, i.e. the spatial/temporal ratio of the area that is covered by a mask on average for a clip. Recall that each mask is constructed by sampling several (possibly overlapping) blocks and taking their union. We change the average

Table 13: **Masking Ratio.** We explore the impact of the spatial and temporal ratio masking. Low spatial or temporal coverage results in a trivial prediction task, which degrades downstream performance.

| Mask Statistics | | 3D Multi-Block Mask Details | | | K400 Acc. |
|---|---|---|---|---|---|
| Avg. Depth | Avg. Spatial Size | Spatial Size of Block | Frames per Block | Blocks per Mask | |
| 100% | 25 % | $112 \times 112$ | 16 | 1 | 0.23 |
| | 50 % | $160 \times 160$ | 16 | 1 | 0.31 |
| | 75 % | $192 \times 192$ | 16 | 1 | 0.44 |
| | 90 % | $176 \times 176$ | 16 | 2 | 0.50 |
| | 95 % | $192 \times 192$ | 16 | 2 | 0.47 |
| 75% | 25 % | $112 \times 112$ | 12 | 1 | 0.10 |
| | 50 % | $160 \times 160$ | 12 | 1 | 0.14 |
| | 75 % | $192 \times 192$ | 12 | 1 | 0.16 |
| | 90 % | $176 \times 176$ | 12 | 2 | 0.22 |
| | 95 % | $192 \times 192$ | 12 | 2 | 0.21 |
| 50% | 25 % | $112 \times 112$ | 8 | 1 | 0.04 |
| | 50 % | $160 \times 160$ | 8 | 1 | 0.11 |
| | 75 % | $192 \times 192$ | 8 | 1 | 0.13 |
| | 90 % | $176 \times 176$ | 8 | 2 | 0.12 |
| | 95 % | $192 \times 192$ | 8 | 2 | 0.14 |

spatial or temporal masking ratio by changing block spatial or temporal size, as well as the overall number of blocks. We found that low spatial or temporal coverage results in a trivial prediction task, which degrades downstream performance. Based on those results, we sample masks that remove roughly 90% of the frame and extend along the entire temporal dimension of the clip by default.

Table 14: **Block Spatial Size.** We investigate the impact of blocks spatial size given an effective masking ratio of 75%. We find that sampling several small blocks to perform better than sampling a single large block.

| Mask Statistics | | 3D Multi-Block Mask Details | | | K400 Acc. |
|---|---|---|---|---|---|
| Avg. Depth | Avg. Spatial Size | Spatial Size of Block | Frames per Block | Blocks per Mask | |
| 100% | 75% | $64 \times 64$ | 16 | 16 | 0.49 |
| | | $96 \times 96$ | 16 | 8 | 0.50 |
| | | $128 \times 128$ | 16 | 6 | 0.49 |
| | | $160 \times 160$ | 16 | 2 | 0.48 |
| | | $192 \times 192$ | 16 | 1 | 0.47 |

In Table 14, we explore different block size given an effective spatial masking ratio of 75% and temporal ratio of 100%. We keep the masking ratio approximately constant by changing the block size and the number of block at the same time. We find that sampling several blocks to perform better than sampling a single large block. Figure 6 visually illustrates the effect of sampling several smaller blocks to construct a mask.

Table 15: **Number of Masks Per Sample.** We explore the effect of sampling several mask for each video clip in the batch. Sampling two masks for each clip, with different spatial block sizes for each, is more effective than sampling just a single mask.

| | 3D Multi-Block Mask Details | | | K400 Acc. |
|---|---|---|---|---|
| Masks per Sample | Spatial Size of Block | Frames per Block | Blocks per Mask | |
| 1 | $160 \times 160$ | 16 | 2 | 0.50 |
| 2 | $160 \times 160$ | 16 | 2 | 0.55 |
| 3 | $160 \times 160$ | 16 | 2 | 0.55 |

In Table 15, we explore the effect of sampling various number of masks per samples. We find that sampling two masks for each clip, with different spatial block sizes for each, to be more effective than sampling just a single mask. We hypothesize that this masking strategy induces complementary tasks. In our experiment, we use this as our default masks sampling.

## D   THEORETICAL MOTIVATION OF EMA FOR L1 LOSS

Consider just the first loss term in the summation of equation 1. To condense the notation, denote the output of the context encoder with parameters $\theta$ by $z_N(\theta)$, and denote the first token output

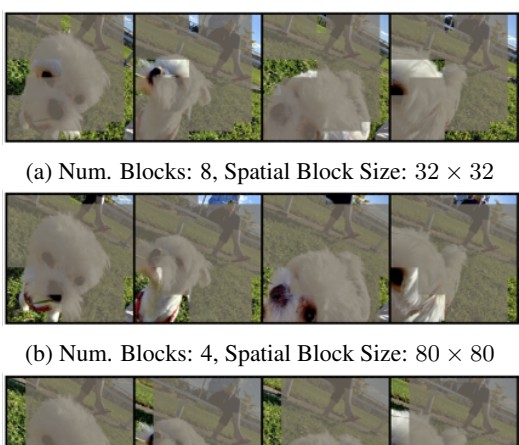

(a) Num. Blocks: 8, Spatial Block Size: $32 \times 32$

(b) Num. Blocks: 4, Spatial Block Size: $80 \times 80$

(c) Num. Blocks: 2, Spatial Block Size: $160 \times 160$

Figure 6: Illustration of mask with number of blocks and block size. Each mask is constructed by sampling several (possibly overlapping) blocks and taking their union.

by the predictor as $p(\boldsymbol{z}_N(\theta)) = [P_\phi(\boldsymbol{z}_N, \boldsymbol{m}_M)]_{i_1}$. Finally, define the corresponding target token (output by the target encoder) as a random vector $X \in \mathbb{R}^d$. Now, if we were to compute the optimal predictor under our loss function, we would obtain the following functional expression,

$$p^\star(\boldsymbol{z}_N(\theta)) = \mathrm{argmin}_p \|p(\boldsymbol{z}_N(\theta)) - X\|_1 = \mathrm{median}(X|\boldsymbol{z}_N(\theta)).$$

Substituting this expression for the optimal predictor into the loss function and evaluating the expected gradient of the context encoder gives

$$\nabla_\theta \mathbb{E}\|p^\star(\boldsymbol{z}_N(\theta)) - X\|_1 = \nabla_\theta \sum_{l=1}^d \mathrm{MAD}(X_l|\boldsymbol{z}_N(\theta)),$$

where $X_l$ is the $l^{\text{th}}$ entry of the random vector $X$, and $\mathrm{MAD}(\cdot\,|\boldsymbol{z}_N(\theta))$ is the median absolute deviation of a random variable conditioned on $\boldsymbol{z}_N(\theta)$. Thus, in the case where the predictor is optimal, the context encoder must learn to capture as much information about the masked clip as possible to minimize the deviation of the target. The hypothesis is that by updating the target encoder weights via an exponential moving average of the context encoder weights, we ensure that the predictor evolves much faster than the target encoder and remains close to optimal, thereby preventing collapse.

