# OpenReview forum: "V-JEPA: Latent Video Prediction for Visual Representation Learning"
_ICLR.cc/2024/Conference — Submitted to ICLR 2024_

### Official Review · Reviewer_V39V · 2023-10-13

**Soundness:** 1 poor
**Presentation:** 2 fair
**Contribution:** 1 poor
**Rating:** 3
**Confidence:** 5

**Summary:**

This paper applies the JEPA architectures to videos by processing 16-frame videos (with a stride of 4) using a 3d-CNN. The model is trained on roughly 2M videos by combining Kinetics 400/600/700, HowTo100M, and SSv2. The authors show that the representation learned after training the encoder performs well on various tasks (video and image tasks).

**Strengths:**

- The biggest strength of this work is the large-scale training on a huge dataset (VideoMix2M), showing the ability of JEPA to scale up.
- This work is evaluated on six different tasks/datasets (3 for videos and 3 for images), which provides a solid baseline for future methods.
- The idea is simple.

**Weaknesses:**

### Novelty

- Using an existing image architecture to process videos by flattening the 3D feature maps into 1D does not qualify as an architectural novelty.
- Videos have different challenges than images, which could have been tackled as novelties, such as how to cover larger temporal context (>2 seconds) or how to capture long-term dependencies (e.g., H-JEPA?)
- For a paper that uses an existing architecture (JEPA), it is expected to see proposals on how to solve challenges of the video domain.

### Experimentation and Comparisons:

- It seems that the experimental protocol is flawed. Neither Table 1 nor Table 2 show the results of the model when trained on the same datasets. This model is trained on more data than all the other models, making a fair comparison impossible. A common approach would be to show the new model results when trained on the same datasets as previous works, then provide the results with the scaled-up dataset (as a plus). It is hard for the reader to validate the model’s performance if more than one thing changes between the rows of a comparison table.
- Also, Figure 1 (Left) shows comparisons, but all models used different sizes and cannot be directly compared.
- Looking through the ablations, the only result I can use for comparison is [Table 3 (b) row 1], [Table 1 row VideoMAE], and [Table 2 row VideoMAE]. All three results use the ViT-L/16 model and pretrain on the same K400 data. Results show that on the Video task for SSv2, VideoMAE outperforms the proposed method by (61.2-50.8=10.4), and on the Image task for IN1k, VideoMAE outperforms V-JEPA by (71.9-66.1=5.8). These results do not show that V-JEPA is SOTA when trained fairly on the same datasets as others.

### Ablations:

- The provided ablations on masking strategy are good, but for a video paper, I expected more ablations on the stride and temporal reception field of the model. It is more likely that the model could perform better if trained on more than 2-second snippets or uses a two-flow architecture (I3D with optical flow?). These ablations are more relevant to video papers.

**Questions:**

- In Paragraph "Comparing with image models, 2nd paragraph": Is it possible that attentive probing improves the performance over linear probing because of the attention mechanism? How does it do with just linear probing?
- Have you considered a multiscale representation, where you concatenate 16 frames with small strides at 224x224 with 16 frames at double the stride at 128x128,  and so on?

---

> ### Author Response · Authors · 2023-11-20
>
> Thank you for your useful feedback. We are glad that you found that we propose a simple idea and solid baselines for future works as well as demonstrate the ability of JEPA to scale.
>
> We address your concern below:
>
> 1) **Novelty:**  We would like to clarify that we don’t claim novelty in terms of model architecture. Rather, this is the first paper to demonstrate that the joint-embedding predictive architecture, which is a learning principle, can be used to learn high-quality representations from video.
>     * V-JEPA is the first approach to learn from video without either pixel reconstruction or an invariance-based learning criteria (contrastive) and without using any text. Previous works exploring this class of learning algorithms (I-JEPA, data2vec, etc.) focused only on Image, speech or text modalities.
>     * To demonstrate the value and impact of this exploration, we evaluate our approach using a ‘frozen evaluation’ protocol as we are interested in video models that can be applied off-the-shelf to many downstream tasks, similarly to models such as DINOv2 or OpenCLIP in the image domain.
>     * In a frozen evaluation setting, V-JEPA achieves (1) the best performance across video models on 6 video and image tasks and (2) outperforms large-scale image-based pretrained models on tasks requiring motion understanding. V-JEPA therefore produces state-of-art video embedding that can be applied off-the-shelf to many video and image tasks without any finetuning.
>     * We agree that developing architectures and methods that further take advantage of the video specificities, such as long duration, is a very important direction for research which is complementary to this work. However, the first step to enable this direction was to demonstrate the suitability of JEPA learning for video, which we have accomplished in this paper.
>
> 2)  **Evaluation:** Thank you for your feedback. To address your concern regarding the evaluation, we pretrained a V-JEPA model with a ViT-L/16 architecture on the K400 dataset only. This experiment uses the same architecture and dataset as the VideoMAE approach allowing for a direct comparison. V-JEPA achieves scores of 78.7, 61.7, and 25.0 on K400, SSv2, and AvA. V-JEPA has a consistent advantage over VideoMAEv1 with an improvement of +0.7, +0.5, and +3.4 points on K400, SSv2, and AvA, respectively.
>
> 3) **Table 3 results:** We apologize for the confusion in the Table 3 ablation experiment. We found a mistake in the reporting of this ablation, which is now corrected. V-JEPA pretrained on K400 achieves a score of 61.7 on SSv2 and 70.4 on ImageNet. While VideoMAE v1 achieves 71.9 on ImageNet (+1.5% compared to V-JEPA pretrained on K400), the V-JEPA K400 model outperforms the identical VideoMAE K400 model on all video tasks.
>
>     Furthermore, V- JEPA ViT-L/16 demonstrates a favorable scaling behavior; by pretraining on VideoMix2M (comprising 2M videos), the performance on ImageNet is boosted by +3.3% top-1. We note that we do not observe a performance improvement when evaluating a VideoMAEv2 model pretrained on the larger UnlabeledHybrid dataset (comprising 1.35M videos).
>
> 4)  **Masking Ablation:** Thank you for your suggestion. We added a new section in Appendix C.4 that ablates different choices regarding the masking strategy. The main findings of this ablation are 1) the importance of high-masking ratio (as highlighted in previous work) 2) masking several blocks performs better than masking one large block, given a fixed masking ratio (recall that each mask is constructed by sampling several (possibly overlapping) blocks and taking their union). 3) sampling several masks with different block sizes for each clip further improves the performance.
>
>  5)  **Linear vs attentive probing:** We perform an extensive analysis on the probing strategies in Appendix C. We found that attentive probing improved the results compared to linear probing for both V-JEPA and the baselines. Using our evaluation protocol, we achieve new state-of-art performance for frozen evaluation using a DINOv2 backbone on three of our evaluation tasks: K400, INaturalist and Places-205 (see Table 9 in Appendix B). Our evaluation protocol also strongly improves the performance of OpenCLIP and VideoMAE models as Tables 8 and 9 show. We therefore choose attentive probing as our default evaluation as it leads to more competitive results among frozen evaluations for all baselines and V-JEPA.
>
> 6) **Multiscale representation:** We agree that exploring multiscale representations with various timescales is a very interesting and promising research direction. We did not explore such an approach in the current work, but plan to do so in future work.
>
> Thank you again for your feedback.  We hope that our answer addresses your concerns and that you will consider raising your score.

---

### Official Review · Reviewer_TgmP · 2023-10-28

**Soundness:** 3 good
**Presentation:** 3 good
**Contribution:** 2 fair
**Rating:** 5
**Confidence:** 5

**Summary:**

In this work, the authors propose a self-supervised mask modeling method from videos. Bascially, the style is like the exsiting mean teacher learning, while the self-distillation is performed for masked regions in the latent space.

**Strengths:**

1 The fundamental training method is crtical for both image and video representation learning.
2 The method seems to be sound.
3 The paper is written well.

**Weaknesses:**

* Methodology

1. Such self-distillation style has been widely used in the literature, which comes from mean teacher [NIPS 2017].

2. Mask modeling has been used in both image and video literatures. Even predicting the latent feature of masked spatio-temporal regions has been recently investigated in UMT [ICCV 2023].

Hence，the combination of 1 and 2 is not quite inspired.

* Results

1. The results show in the abs and Figure 1 are not quite unfair actually, since the model sizes are not the same.

2. The pretraining data is VideoMix2M, which are different from others. It is a bit doubtful about whether the improvement comes from the self-supervised method.

3. It would be also great to show the finetuning results on the video benchmarks (e.g., K400, SthV2) for SOTA comparison.

**Questions:**

Please see weakness for reference.

---

> ### Author Response · Authors · 2023-11-20
>
> Thank you for your feedback. We agree with your statement that ‘fundamental training method is critical for images and video representations learning’ and we are glad that you found our method sound and our paper well-written.
>
> Regarding your concerns:
>
> 1) **Novelty:** Our main contribution is demonstrating that a joint-embedding predictive architecture can successfully and effectively be used for self-supervised representation learning from video.
>
>     Additionally, one of our main technical innovations is to learn both video representations and the target representations simultaneously, without any form of supervision. Methods like UMT require a pre-trained teacher to compute the target representation. In the case of UMT, the teacher is a frozen vision-language model pretrained on a large corpus of image-text pairs, and hence benefitting from the textual supervision. In contrast, our approach only uses video data for pretraining, and does not require any text-paired data or other pre-trained models.
>
> 2) **Model sizes and datasets:** Thank you for your suggestion. We agree that both the VideoMix2M dataset and proposed V-JEPA methodology contribute to the V-JEPA results shown in Table 1. To disentangle some differences from previous work, we ran an additional experiment where we pretrain a ViT-L/16 on only the K400 dataset (comprising 240K videos) using the same pretraining hyperparameters described in Section 4.1. We compare this model to a VideoMAEv1 ViT-L/16 which was also pretrained only on K400. V-JEPA achieves scores of 78.7, 61.7, and 25.0 on K400, SSv2, and AvA. V-JEPA has a consistent advantage over VideoMAEv1 with an improvement of +0.7, +0.5, and +3.4 points on K400, SSv2, and AvA, respectively.
>
>     Furthermore, our V-JEPA approach demonstrates favorable scaling behavior: by pretraining on VideoMix2M (comprising 2M videos), the performance is boosted by +0.5, +5.3, and +0.6 points, respectively. We note that we do not observe a similar improvement when evaluating the VideoMAEv2 model, which was pre-trained on a dataset of 1.35M videos (including a private subset, preventing us from training V-JEPA models in the same setting).
>
> 3) **Finetuning:** Please note that finetuning results were reported in Appendix C.2. We have included these in the main paper, and also included a comparison with VideoMAE v1. We find V-JEPA to outperform VideoMAEv1 finetuning by +0.1% on K400 and +0.5% on SSv2, while also training for only ~60% of the VideoMAEv1 training time. The VideoMAEv2 model improves on VideoMAEv1 by training a ViT-g/14 model for roughly an order of magnitude more iterations. Efficiency is maintained by using a shallower 4-layer decoder and employing aggressive decoder masking. This VideoMAEv2 ViT-g/14 model outperforms our V-JEPA model in finetuning by +0.5% on K400 and +1.7% on SSv2; however, this result requires training 6.5x longer than our baseline V-JEPA model.
>
>
> Thank you again for reviewing our paper and providing constructive feedback. We hope that our responses have addressed your concerns and that you will consider increasing your score.

---

### Official Review · Reviewer_eUQX · 2023-11-08

**Soundness:** 3 good
**Presentation:** 2 fair
**Contribution:** 3 good
**Rating:** 5
**Confidence:** 4

**Summary:**

This paper studies the video representation learning via predicting the latent representation of video patches. The argument is that by predicting the latent representation, the model will achieve better results than the model trained via Masked AutoEncoder style training. By comparing with other MAE style approaches, the proposed one achieve higher accuracies on Kinetics, SSv2, and AVA dataset via simple linear probing tuning. The model also achieves quite significant performance on the image recognition. Those results show the proposed approach is quite good for representation learning.

**Strengths:**

The paper idea is quite simple and straight forward. It shares the same high-level idea as the MAE. While the MAE focuses on predicting the patch, the proposed idea focuses on predicting the embedding of the patch.

The proposed approach achieves quite significant performance boost on all the datasets, which is a little bit surprising. As the proposed approach is very similar to MAE, I wonder the reason behind the performance boost.

If the proposed approach and its performance can be verified, then it might be a quite good paper for the community to have.

**Weaknesses:**

The paper writing should be improved. The methodology part is quite hard to read. The high-level idea (sect. 3.1) is okayish to understand. But the application to video part (sect. 3.2) is not. Concretely, as the input has already been processed by the 3D ConvNet as a 1D feature, I don't know how to obtain the 3D sin-cos positional embedding (mentioned in Predictor section)?

After reading the paper, I still don't know how to obtain the self-supervised video features. Is it the output of the target encoder or the output of the context encoder?

For Table1, it is known that joint training on SSv2 and Kinetics will improve both dataset's performance as mentioned in (Zhang, B., Yu, J., Fifty, C., Han, W., Dai, A. M., Pang, R., & Sha, F. (2021). Co-training transformer with videos and images improves action recognition. arXiv preprint arXiv:2112.07175) For a fair comparison with VideoMAE, the proposed approach should be pretrained on the Kinetics400 dataset.

**Questions:**

Please address the questions listed in the weakness.

In addition to that, I wonder how to do the inference? Which part of the model (context encoder, predictor, or the target encoder) are included in the inference process?

Why the model could achieve much better performacne than the VideoMAE? Is it because of the mixture video datasets for pretraining (if so, the whole paper might not be that exciting)? If predicting the embedding is better than predicting the patch, could you provide a conceptual reasoning for that?

Before reading Table 1 carefully, I think this paper is quite good. However, after reading Table 1 and comparing VideoMAE and V-JEPA on the ViT-L/16, I am afraid the main performance improvement is not brought by the proposed approach, but by the mixture of video datasets for pretraining.

---

> ### Author Response · Authors · 2023-11-20
>
> Thank you for your valuable feedback! We are glad that you found the VJEPA idea simple and that it achieves a significant performance boost.
>
> We address your concern below.
>
> 1) **3D positional embedding:** Thank you for your feedback. We added a new figure (Figure 5 in appendix) and have reworked the methodology section to clarify this point. Specifically, to convert a video clip of size 16x224x224x3 into a 1D token sequence, we apply a 3D convolution comprising d filters of size 2x16x16 with a temporal stride of 2 and a spatial stride of 16, resulting in a tensor of shape 8x14x14xd. Next, we add absolute 3D sin-cos positional embeddings to the spatio-temporal feature map and flatten it, resulting in a 1D token sequence of shape 1568xd.
>
> 2) **SSL feature:** The self-supervised learning features are the output of the target encoder, which is then fed to an attentive or linear probe. Please refer to appendix B.1 for more details
>
> 3) **Comparison on K400:** Thank you for your suggestion. We added an experiment where we pretrain a ViT-L/16 using V-JEPA on only the K400 dataset (comprising 240K videos) using the same pretraining hyperparameters described in Section 4.1.  We compare this model to a VideoMAE ViT-L/16 also pretrained only on K400. V-JEPA achieves scores of 78.7, 61.7 and 25.0 on K400, SSv2 and AvA. V-JEPA has a consistent advantage over VideoMAE with an improvement of +0.7, +0.5 and +3.4 points on K400, SSv2 and AvA.
>
>     Furthermore, V-JEPA ViT-L/16 demonstrates a favorable scaling behavior; by pretraining on VideoMix2M (comprising 2M videos), the performance is boosted by +0.5, +5.3 and +0.6 points, respectively. Both the VideoMix-2M dataset and the V-JEPA approach therefore contribute to the results. We note that we do not observe a similar performance improvement when evaluating a VideoMAEv2 model that was pretrained on the UnlabeledHybrid dataset (comprising 1.35M videos, including a private subset).
>
> 4) **Importance of latent-space prediction:** By predicting in latent space, our model can potentially eliminate unnecessary pixel-level details when encoding a video with the target encoder. As a result, the model has the flexibility to focus only on predictive features from the videos and discard irrelevant information from the target representation. In contrast, a reconstruction based approach needs to model every input pixel correctly to satisfy its loss.
>
> Thank you again for your feedback! We hope that our answer addresses your concerns and that you will consider raising your score.

---

### Official Review · Reviewer_zTBB · 2023-11-08

**Soundness:** 3 good
**Presentation:** 3 good
**Contribution:** 2 fair
**Rating:** 6
**Confidence:** 3

**Summary:**

The paper introduces VJEPA, a representation learning based on joint embedding predictive architecture (JEPA) to learn from video data. VJEPA is a self-distillation method that predicts embeddings (latents) of masked out regions of the input conditioned on embeddings of visible patches from a context encoder. The predicted latents are compared to latents from a teacher network whose weights are updated in via an exponential moving average (EMA) operation. The paper proposes a new dataset called ``VideoMix2M'' to train VJEPA. Empirical evaluations are performed to show the effectiveness of VJEPA over other video-based and image-based methods for several downstream tasks.

**Strengths:**

- The paper is a nicely executed empirical study that extends I-JEPA [Assran et al, 2023) to video data
- The main message for me as a user is that (pre)training on video data can be fast by decoding/predicting in latent space. This is not a big surprise given that VJEPA inherits the strengths of I-JEPA. However, the paper shows empirically that there may not be a need need to sacrifice performance for speed
- Empirical evaluations with ``frozen'' backbones (encoders) are conducted with academic video and standard image benchmarks that suggest that representations on video work better on video data while being competitive on image benchmarks

**Weaknesses:**

- The method proposed in the paper is an extension of I-JEPA to video. The approach used to implement masking across the temporal dimension (``3D-multiblock'') and latent space prediction borrow heavily from I-JEPA. The novelty of the method introduced here is very limited for the reason mentioned above

- A key component that determines the success of VJPEA is multiblock design. I would have liked to have seen careful ablation of the design choices of this block including the scale factors and also aspect ratios used to design the mask

- The method uses an evaluation scheme (attentive probing) which may be the way to probe representations in foundation models. The biggest problem I see with this scheme is that it may make it hard for readers to compare performance across publications.

Also, please note my questions in the **Questions** section

**Questions:**

-  Potential missing background/related work: the paper "Siamese Masked Encoders" (https://arxiv.org/abs/2305.1434)
- Did you conduct a full ablation of all the design hyperparameters used in multiblock. If so, please this ablation in the appendix as done in I-JEPA for instance
- Table 3 ablation - is randomly picking images from a video sequence the best approach to create a training dataset for image-based backbones. Also, the backbones used for VJEPA and I-JEPA in Table 3(a) are different. If results are available for the same backbone please consider including those t help a reader understand the design choices better.

---

> ### Author Response · Authors · 2023-11-20
>
> Thank you for your review. We are glad that you found our work to be a nicely executed empirical study. We address your comments below.
>
> 1) **Novelty:**  V-JEPA is the first paper to demonstrate that self-supervised learning with a joint-embedding predictive architecture works for video. Previous works have only explored joint-embedding predictive architectures with images, text, or speech modalities.
>
> 2) **Masking Ablation:** Thank you for your suggestion. We added a new section in Appendix C.4 that ablates different choices regarding the masking strategy. The main findings of this ablation are 1) the importance of high-masking ratio (as highlighted in previous work) 2) masking several blocks performs better than masking one large block, given a fixed masking ratio (recall that each mask is constructed by sampling several (possibly overlapping) blocks and taking their union). 3) sampling several masks with different block sizes for each clip further improves the performance.
>
> 3)  **Attentive probe comparison with previous work:** For the initial submission we ran the attentive probe protocol with open-source baseline models, and also reported attentive probing results from the VideoGLUE paper, which provides an extensive evaluation of previous video models under frozen evaluation. Previous work VideoMAE v1 reports frozen evaluation with a linear probe on SSv2 (38.9% top-1). We significantly improve upon their reported numbers by evaluating their model using an attentive probe (61.2% top-1 with ViT-L/16). To ease comparison with previous work, we also report fine-tuning evaluations, which we moved from the appendix to the main paper in the revision.
>
> 3)  **Siamese Masked Encoder:** We have cited Siamese Masked Encoders and discussed this interesting work in the main paper. SME uses a frame-level encoder, and trains a decoder to reconstruct missing pixels in future frames. By contrast, V-JEPA uses a video encoder and does not require reconstructing pixels, but predicts in latent space. It would  be interesting to combine the cross-attentive architecture of Siamese Masked Autoencoder with a JEPA loss in future work!
>
> 4)  **Ablation Table 3:** In this ablation, V-JEPA and I-JEPA see the same exact training data (videos) in the same exact sequence. However, because I-JEPA can only process individual frames, we can only compute the I-JEPA loss on individual frames. We did not have enough time to train a smaller I-JEPA backbone on K400, but we are planning to include this additional ablation and finer-grained comparison in the paper.
>
> We thank the reviewer for their feedback, and for their time in reviewing our work!

---

> > ### Comment · Reviewer_zTBB · 2023-11-22
> > **Rebuttal response**
> >
> > I thank the authors for their rebuttal. The new results on masking strategy ablation is likely to be useful to the community in the same way the ablation done in I-JEPA is useful for the researchers and users of I-JEPA. I agree with other reviewers that the contribution in this paper is not the method itself but that it clearly points out one way to apply JEPA to video data. I will keep my score unchanged for now.

---

### Official Review · Reviewer_HoWZ · 2023-11-10

**Soundness:** 1 poor
**Presentation:** 2 fair
**Contribution:** 1 poor
**Rating:** 3
**Confidence:** 4

**Summary:**

This work extends I-JEPA to videos for self-supervised video pretraining. Models of varying scale are trained using a 2 million video custom mix of common datasets (Kinetics-400/600/700, HowTo100M, SSv2). The resulting frozen models are compared to some prior work in literature (e.g. VideoMAE, InternVideo) and shown to have better performance on classification and action recognition.

**Strengths:**

1. As an extension of Data2Vec, I-JEPA etc class of work to videos, it is a useful baseline for the community to know.
2. The experiments and methodology are clear and easy to understand.

**Weaknesses:**

1. As a extension of I-JEPA to videos, technical novelty is extremely limited.
2. A much bigger issue is that evaluations are problematic, by either not comparing to the relevant work or by comparing to prior work in a misleading way, contrary to the evaluation protocols the whole community has been using.

For example:
a) Most recent related work has adopted a finetuning evaluation protocol (e.g. VideoMAE, VideoMAEv2, ST-MAE, Hiera, MaskFeat, OmniMAE, MCMAE) and yet such evaluation is almost entirely absent from this work (except for highly limited comparison in Table 11, where V-JEPA performs worse than prior work despite a misleading comparison*), portraying a highly misleading picture of not only the state of the literature but also of the capabilities of the V-JEPA models in comparison. It is well known since even the original MAE work that such models do not have the best performance when frozen, but even finetuning 1 or a small number of layers makes a large difference, demonstrating how frozen evaluation can be quite misleading. Classification aside, on action recognition, the gap between frozen and finetuning can be particularly large, which can again be quite misleading.



Also, notably missing are references/discussion and comparison to some of the stronger recent work (e.g. UnMasked Teacher, Li et al 2023; Hiera, Ryali et al 2023; MCMAE, Gao et al, 2022, etc)

b) Transfer to images is also explored, but once again, comparison to prior work is problematic, e.g. notably missing is any comparison to VITO,  Parthasarathy et al, 2023.

c) The sample efficiency discussion is problematic in several ways, e.g. numbers of images are being compared to number of videos (Table 12). Another example is the claim that training schedule is "shorter" in terms of numbers epochs and therefore more efficient. It is is not particularly meaningful to compare number of epochs. It is far more meaningful for example, to compare wall clock training time for a comparable accuracy with a recent work, e.g. VideoMAEv2, Wang et al, 2023.

*VideoMAEv1 achieves similar performance with much less data and model size as v2 on the considered metrics, so the speedup in Table 11 is likely in large part an artifact of the larger amount of data that particular model (v2, ViT-g/14) was trained on.

**Questions:**

It would be great if the authors can address the discussed limitations/weaknesses.

---

> ### Author Response · Authors · 2023-11-20
>
> Thank you for your comments and useful feedback. We are glad that you found our work provides a useful baseline for the community to know, and that the methodology and experiments are clear and easy to understand. We address your concerns below:
>
> 1) **Novelty:**  V-JEPA is the first model to learn from video using a Joint-Embedding Predictive Architecture. As you have pointed out, previous works exploring a similar class of algorithms (I-JEPA, data2vec, etc.) only focused only on image, speech or text modalities.
>
> 2) **Frozen evaluation:** Our motivation is to study the ability to produce video embeddings for off-the-shelf use in downstream tasks, and thus we respectfully disagree that the frozen evaluation protocol is not interesting or misleading. In fact, the frozen evaluation protocol is standard in computer vision, such as in self-supervised learning with images or text. While it is true that the video community has mostly focused on end-to-end fine-tuning, there are many notable works that also explore frozen evaluation of video models/tasks, such as VideoGLUE (Yuan et al., 2023), VITO (Parthasarathy et al., 2022), DINOv2 (Oquab et al., 2023), Unsupervised Video CNNs (Wang & Gupta, 2015), etc. In fact, even the VideoMAE v1 mentioned in the review reports frozen evaluation with a linear probe on SSv2 (38.9% top-1). We significantly improve upon their reported numbers by evaluating their model using an attentive probe (61.2% top-1 with ViT-L/16). We also improve the reported DINOv2 frozen evaluation number on K400 from 74.4 to 84.4 top-1 by using an attentive probe.
>
>     Having said that, we hope to address your concern and have updated the main paper to state explicitly that absolute state-of-art is obtained by models that are fine-tuned on each task individually and report the best performances on video tasks. We are happy to revise our paper further to clarify those points.
>
> 3) **Finetuning evaluation:** We thank the reviewer for their suggestion, and have added the VideoMAEv1 baseline in our Finetuning Evaluation table. While VideoMAEv1 uses a smaller model than VideoMAE v2 and is trained with far fewer samples, its cost per iteration is higher than VideoMAEv2, since v2 uses a much shallower decoder and aggressive decoder masking. We find V-JEPA to outperform VideoMAEv1 finetuning by +0.1% on K400 and +0.5% on SSv2, while also training for only ~60% of the VideoMAEv1 training time.
>
> 4) **Other papers:**  Thank you for pointing out those works, we have added citations to these papers and discussed them in the main text as well. We have also
> a) added the suggested evaluation of Hiera in Table 1. A V-JEPA L/16 trained on K400 outperforms a bigger Hiera-H model (also trained on K400) by +5.4% on K400, +1.0% on SSv2, and +7.5% on AVA.
> b) We cite and discuss VITO in the main paper. Note that the VITO video model reports a frozen evaluation on ImageNet of 66.2 with a ResNet50 and a linear probe. Their model is not publicly available so we cannot validate this result or rerun the frozen evaluation, but our smallest V-JEPA model, a ViT-L/16 with an attentive probe, achieves a top-1 of 73.7 , which is +7.5% higher.
>
> ** We report wall-clock comparisons in Table 4 in the main paper.
>
> Thank you for all your comments, which we believe have strengthened our work; we ask that you consider increasing your score to reflect the revisions and clarifications.

---

> ### Comment · Reviewer_HoWZ · 2023-11-22
> **Response**
>
> Thank you for your response.
>
> a) To clarify, my concern is not that linear evaluation is _uninteresting_. Indeed, as you mention many works use (e.g. SimCLR, BYOL, VICReg, DINO, DINOv2, BYOL, etc) use a linear evaluation protocol. In these cases, the pretraining task of instance discrimination and use of an EMA teacher makes these methods perform very well under linear evaluation. However, it is _misleading_ for MIM works and particularly so when not using an EMA teacher. This is well known since the original MAE paper. Moreover, as the video community has primarily adopted finetuning as the standard evaluation protocol, reporting evaluation almost entirely on linear evaluation does not give the reader a clear picture and does not situate V-JEPA in the literature. So - it is great V-JEPA reports linear evaluation. And great to see Table 3 now reports some finetuning results, but it is quite limited. For e.g. UMT (as also mentioned by another reviewer) is a highly relevant and strong baseline for both linear and finetuning evaluation and it is not clear to me why it is missing. Yes, this used a CLIP teacher and thus benefits from language supervision, but this is also true of OpenCLIP. Why is OpenCLIP a valid baseline, but not UMT?
>
> b) There is a related aspect I find myself confused about. It is claimed that focus is frozen evaluation for downstream tasks. Maybe it is just me, but why is this setting important and what tasks is it relevant for? In what settings is it feasible to train an attention head for a downstream task but infeasible to finetune a small number of layers of the model (or additional new layers) ? This is particularly confusing when so many effective adaptation or finetuning strategies exist in literature.
>
> c) Moreover, many video papers typically report FLOPs or inference speed to enable more system level comparisons. For example, in Table 2,3 you compare models of various sizes (also an issue in Figure 1) and using clips of different resolution yet comparison is difficult, e.g. Table 3 V-JEPA uses H-384 while comparing to VideoMAEv1/v2 at 224 resolution. It is great you report training speed up. Reporting inference speed or FLOPs (as is common in related work) would make the system level comparison much more helpful.

---

> > ### Author Response · Authors · 2023-11-23
> >
> > 1) Thank you for your response. We clarify the points below
> >     * We would like to clarify that V-JEPA is a MIM approach, as it only relies on masking to learn representations; the approach does not perform instance discrimination. Indeed, one of the contributions in this paper is to show that it is possible to learn competitive MIM video representations for frozen eval using only masking, which as you have indicated is a novel observation.﻿ ﻿ Additionally, we now report both frozen evaluations and fine-tuning performances for VideoMAE and VideoMAEv2 in the main paper to give a complete view of previous MIM empirical performances.
> >     * We completely agree that UMT is a highly interesting approach relying on pretrained image and text encoders, but UMT was only published at the ICCV2023 conference, which was after the ICLR submission deadline, and we were not aware of this work before its publication. Unfortunately we did not have time to run the evals on their models during the rebuttal period but we will make sure to include these evaluations in the next revision. However, please note that we do currently report the state-of-the-art models for all the video tasks at the beginning of section 4.2.
> >
> > 2) Indeed we are happy to clarify this point. There are many practical situations of interest, especially in industry, where there is a single backbone processing visual data that is intended to serve many downstream task-specific models. This is also popular in cloud computing APIs, where a single model endpoint is exposed to edge users. The frozen-evaluation is thus the precise protocol for evaluating models in this setting. Examples of commercial cloud systems providing embedding APIs include:
> >     * Google Vertex AI: image embeddings & text embeddings
> >     * Amazon Bedrock: text embeddings
> >     * OpenAI Embeddings: text embeddings
> >     * Cohere Embed: text embeddings
> >     * Hugging Face Image Embedding Inference API: image embeddings
> >
> > 3) As you point out, we do report wall-clock-time for training models and baselines (e.g., Table 3), and we will be sure to include FLOPs for inference speed in the paper as well!
> >
> > We wish to thank you for your insightful questions, and we hope that you do find the value in the results presented in this work for the community and consider increasing your score.

---

> ### Comment · Reviewer_HoWZ · 2023-11-23
>
> Thanks for your response.
>
> 1. Yes, I don't disagree that V-JEPA is an MIM approach. It is also an instance discrimination approach where you can consider the masking as an augmentation - this indeed why I mention in my response that primarily focusing on linear evaluation of MIM methods that do not use an EMA teacher is particularly an issue. In fact, this is not novel, e.g. MSN (Assran et al) and iBOT (Zhou et al). Using an EMA teacher with MIM also is known to help in finetuning setting, e.g. data2vec, data2vec2.0 (Baevski et al). Despite limited novelty - I still think it is could be useful as a baseline for the community to have a study that takes similar approach for video - but doing so necessitates fair comparison with prior work in the standard protocol. I would encourage the authors to expand Table 3 to properly situate V-JEPA vs prior work - this does not need any additional work or experiments. It will also be useful to have a similar finetuning comparison on AVA.
>
> 2. Thanks for the response regarding practical situations of interest. Unfortunately this still leaves me confused. If we adopt this perspective and assume V-JEPA is behind an API. It is not the embeddings that are being directly evaluated though. A head consisting of a single layer is being trained with data augmentations. Why is this more feasible to train than a head with more layers?
>
> 3. Thank you, adding FLOPs will be very helpful.

---

> ### Author Response · Authors · 2023-11-23
>
> Dear Reviewer,
>
> 1) We respectfully disagree with some of the statements in the first point. Please note:
>     * There is absolutely no instance discrimination component in V-JEPA. Instance discrimination refers to training a network to be able to classify individual instances in the dataset, i.e., each instance has its own class. This can be implemented parametrically (Dosovitskiy et al., 2014) or non-parametrically (Wu et al., 2018). Contrastive learning methods, such SimCLR or PIRL also fall within the instance-discrimination framework since they try to classify the positive instance (an augmented view of the sample) amongst a group of negatives. The V-JEPA loss trains a network to predict missing regions in a masked video; the method does not try to discriminate between different videos in the dataset and there is no coupling between samples in the dataset.
>     * V-JEPA is not learning invariance to masking, in contrast to image methods such as MSN which learn invariance to random resized crop, missing patches, and color augmentations. Since there are none of these geometric or photometric augmentations in the V-JEPA pipeline, learning invariance to missing patches would result in a collapsed representation. As you can see in the MSN paper, Table 9, when using only masking augmentations, MSN achieves only 7% top-1 accuracy on ImageNet. Whereas V-JEPA achieves 77.9% top-1 frozen evaluation accuracy on ImageNet with only masked pretraining on a video dataset. We will update the paper to include visualizations decoding the output of the V-JEPA predictor to demonstrate this point. The closest works to V-JEPA are data2vec, I-JEPA, and data2vec 2.0, and these works only explore image, speech, and text modalities, not video.
>
> 2) Yes this is a good point, and in fact we have tried training deeper probes on top of the frozen network, but this does not improve performance compared to a single layer attentive probe. We will include this ablation in the appendix!
>
> We hope this clarifies any confusion, and we sincerely request that you increase your score in light of these points. We believe V-JEPA is a valuable baseline for the community, and the first method to demonstrate this learning principle for video.

---

> ### Comment · Reviewer_HoWZ · 2023-11-23
>
> Thanks for your response.
>
> Respectfully, I do not agree that V/I-JEPA, data2vec, data2vec2.0 etc are not instance discrimination. A method can be explicitly instance (or-sub instance) discrimination like (e.g. SimCLR) or not (e.g. BYOL). But it is alright if we disagree on this, it does not matter if we call it instance discrimination or not. As I have already mentioned, focusing almost entirely on linear evaluation of MIM based methods that _do not_ use an EMA teacher, regardless of whether we call it instance discrimination or not presents a misleading picture. Adding Table 3 helps, but still quite limited.
>
> It would also be helpful if an expanded discussion of why frozen evaluation is the focus is included in the paper. In my humble opinion, "so that it can be used behind an API which does not offer finetuning services" seems somewhat contrived and limiting; maybe that is just me. But it seems to me that this should atleast be mentioned upfront as the setting of interest. I'll leave it to other reviewers to decide if they find this setting to be of wide enough interest.

---

> ### Author Response · Authors · 2023-11-23
>
> We appreciate your consideration of this work and we agree with your points regrading linear probes. This is why we have adopted the more flexible attentive probe so as to produce the strongest baselines, as demonstrated in the attentive probe vs linear probe ablation in Appendix C.1.
>
> Please note the following points:
> * All frozen evaluations in the main paper are reported with an attentive probe, which has been show to lead MIM methods to be to highly competitive in frozen evaluation with invariance-based pretraining on images, as demonstrated in CAE (Chen et al., 2023).
> * Secondly, we find that MIM methods, such as VideoMAE, actually outperform the instance-discrimination approaches with frozen evaluation. For instance, VideoMAE outperforms InternVideo in frozen evaluation on all considered video tasks (K400, SSv2, AVA), and outperforms VATT on K400 and SSv2, which also relies on an instance discrimination loss. Please refer to Table 1.
>
> * With regards to expanding the fine-tuning results in  Table 3, we will certainly add the suggested UMT baseline, include FLOPS for inference, and incorporate additional baselines. However, please note that Table 3 does already compare with the state-of-the-art methods on these evaluations.
> * While we do currently motivate the focus on frozen evaluation on Page 3, we will be sure to expand this discussion. There are several recent works in video advocating for the frozen evaluation under the attentive protocol, such as VideoGLUE, and we will be sure to incorporate some of this discussion as well.
>
> We hope that these changes address your concerns and kindly ask if you believe these changes would be sufficient, or if you would expect any other revisions to the paper?

---

> ### Comment · Reviewer_HoWZ · 2023-11-23
>
> I apologize, I misspoke. I meant to say _frozen_ evaluation instead of linear evaluation - thanks for bringing this up. I'm glad we agree on the limitations of linear probing. I can also agree that attentive probing is generally a better choice for MIM methods not using an EMA teacher, though less perhaps less appropriate than finetuning (even a small number of layers). You mention VideoGLUE advocating for frozen evaluation - I'm confused by this. One of the important messages from VideoGLUE is that it is important to evaluate under more than one protocol, the picture could look very different (e.g. Figure 3, left from VideoGLUE).  This is not inconsistent with one of the main concerns I have been expressing - that focusing almost entirely on frozen evaluation can be misleading. Toward this end, expanding Table 3 will be very helpful, thank you. Expanding to include even just the methods you have currently in Table 1 and UMT would already be much better*.
>
> I did notice the discussion on page 3 regarding frozen evaluation, but the stated goal there is to have a general video model that can be quickly adapted to a wide variety of tasks. Adaptors or tuning a small number of layers are also feasible options toward this goal and can often be far more appropriate for methods like VideoMAE - cheap and most of the model can stay general purpose. Even beyond that methods like CoCa for e.g. could be more performant and efficient with adaptors (as shown in VideoGLUE).
>
> Concretely, as the paper currently stands, even with an updated Table 3, it is unclear when/why** for example a practitioner should use a V-JEPA model/approach. If they were willing/able to use adaptors or finetune a small number of layers or do e2e finetuning, there does not appear to be a good reason to use V-JEPA based on experiments presented in the paper? I do agree with you that if the goal is to provide an API with no finetuning services, then this is the best evaluation protocol. It seems to me this is quite limited and narrow application.
>
>
> *I do not follow how Table 3 already compares to state of the art as you say, given e.g. UMT, ViT-e, ViT-22B etc can achieve better performance (this is not a criticism, I am just confused since VideoMAEv1/v2 is not the state of the art?)
>
> ** Faster training time could a reason of course, but the faster convergence of such approaches is already known from prior works like data2vec2.0, I-JEPA etc. Certainly there is utility to verifying this for video, but I'm sure we can agree that is a much more limited contribution.

---

> ### Author Response · Authors · 2023-11-23
>
> * Clarification about table 3:  We already report state-of-art performance for K400 at the beginning of section 4.2. We are happy to add them to Table 3.
> * Adaptation protocol: We agree that it might be interesting to explore different protocols. In this work, we focus on learning a probe on top of a frozen backbone as it is a standard practice in the SSL community to evaluate the quality of a representation.
>
>      We note that our results using an attentive probe outperforms the low-rank adapator results reported in VideoGLUE .
>
> Thank you for the discussion.

---

### Author Response · Authors · 2023-11-20
**Response to All Reviewers**

Thank you for taking the time to review our paper and for the useful feedback. We are glad that reviewers found our paper proposed a simple idea (**Reviewer TgmP, eUQX**) with nicely executed empirical studies (**Reviewer zTBB, V39V**) and achieves ‘quite significant performances’ (**Reviewer eUQX**) ‘useful for the community to know’ (**Reviewer HoWZ**).

**Novelty:**
We note that V-JEPA is the first work to demonstrate the principle of learning with a joint-embedding predictive architecture from video, i.e., without using either pixel reconstruction, invariance-based learning criteria (contrastive), or distillation from pretrained image/text models. Previous works exploring this class of algorithms (I-JEPA, data2vec, etc.) focused only on image, speech or text modalities.

**Impact:**
Our goal is to develop strong video models which produce embeddings that can be applied off-the-shelf to many downstream tasks, similarly to generalist models such as DINOv2 or OpenCLIP in the image domain, thus, we extensively evaluate our approach using a frozen evaluation protocol. Frozen evaluation is a standard evaluation protocol that has been thoroughly explored in the self-supervised learning literature in order to explore the quality of the learned representations (SimCLR, BYOL, VICReg, DINO, DINOv2, VideoGLUE, VITO, \rhoBYOL, VideoMAE…)

We demonstrate that our approach achieves (1) the best performance across video models on 6 video and image tasks in frozen evaluation (2) significantly outperforms large-scale image-based pretrained models on tasks requiring motion understanding, demonstrating the value of video pretraining.

We have addressed reviewer specific comments individually, and wish to thank all reviews for their useful feedback, and for taking the time to review our work. Specifically, we have
* Included a V-JEPA pretraining on K400 for direct comparison with VideoMAE
* Reorganized methodology section
* Added extended masking ablation in the appendix
* Moved and broadened fine-tuning comparisons in the main paper

Please refer to the update paper to see the revisions in RED.

---

### Meta-Review · Area_Chair_EakB · 2023-12-09

**Metareview:**

The paper introduces an extension of I-JEPA to video, called V-JEPA. The approach, in a nutshell, deals with self-supervised learning of video representations appropriate for video understanding. In particular, the method learns to predict masked regions in representation space.

The reviewers agree that the paper addresses an important problem of video understanding. And does so in a natural way by extending an existing proven method.

However, all reviewers find that the propose extension is a straightforward application of I-JEPA to video and lacks novelty. In particular, as reviewer V39V points out, that application misses the point of trying to address some of the unique modeling challenges that video data, such as temporal correlations due to motion.

More importantly, the presented results aren't strong enough to compensate for lack of algorithmic novelty. In particular, as two reviewers point out, the evaluation is focused on linear probing, which despite being important, isn't always the way how these models are used and is usually inferior to fine-tuning. The authors provide fine-tuning results which are marginally better to SOTA and as such many of the reviewers do not consider convincing.

**Justification For Why Not Higher Score:**

The reviews are 2 x reject, 2 x borderline reject and 1 x borderline accept. All reviewers cite limited novelty, and many of the reviewers find issues with the limited evaluation. The only borderline accept review is not impressed with the results and cites limited novelty as well. As such, unfortunately the paper is rejected from ICLR 2024.

**Justification For Why Not Lower Score:**

N/A

---

### Decision · Program_Chairs · 2024-01-16

Reject